# Cellular-resolution mapping uncovers spatial adaptive filtering at the rat cerebellum input stage

Stefano Casali[1,5], Marialuisa Tognolina[1,5], Daniela Gandolfi[2], Jonathan Mapelli [2,3] & Egidio D'Angelo [1,4✉]

Long-term synaptic plasticity is thought to provide the substrate for adaptive computation in brain circuits but very little is known about its spatiotemporal organization. Here, we combined multi-spot two-photon laser microscopy in rat cerebellar slices with realistic modeling to map the distribution of plasticity in multi-neuronal units of the cerebellar granular layer. The units, composed by ~300 neurons activated by ~50 mossy fiber glomeruli, showed long-term potentiation concentrated in the core and long-term depression in the periphery. This plasticity was effectively accounted for by an NMDA receptor and calcium-dependent induction rule and was regulated by the inhibitory Golgi cell loops. Long-term synaptic plasticity created effective spatial filters tuning the time-delay and gain of spike retransmission at the cerebellum input stage and provided a plausible basis for the spatiotemporal recoding of input spike patterns anticipated by the motor learning theory.

[1] Department of Brain and Behavioral Sciences, University of Pavia, I-27100 Pavia, Italy. [2] Department of Biomedical, Metabolic and Neural Sciences, University of Modena and Reggio Emilia, I-41125 Modena, Italy. [3] Center for Neuroscience and Neurotechnology, University of Modena and Reggio Emilia, I-41125 Modena, Italy. [4] Brain Connectivity Center, IRCCS Mondino Foundation, I-27100 Pavia, Italy. [5]These authors contributed equally: Stefano Casali, Marialuisa Tognolina. ✉email: dangelo@unipv.it

Long-term synaptic plasticity, in the form of potentiation or depression (LTP and LTD, respectively), is thought to provide the substrate for adaptive computation in brain circuits. LTP and LTD are usually investigated as a single synapse phenomenon under the assumption that multiple independent changes will eventually shape local field responses and brain computation[1–4]. In fact, individual neurons are integrated into local microcircuits, which form effective computational units such as cortical microcolumns[5] or cerebellar microzones[6–8]. It is therefore conceivable that, like neuronal activity, LTP and LTD are also spatially coordinated within local neuronal assemblies that we will define as units. However, the fine-grained organization of changes in such units remains largely unknown despite the potential impact they might have on microcircuit computation.

The cerebellum has long been thought to perform two main operations, i.e., spatiotemporal recoding of input signals in the granular layer followed by pattern recognition in the molecular layer[9–11]. The nature of these operations has been investigated in some detail, showing that single granule cells can actually regulate the gain and timing of spike emission under the control of the local inhibitory network formed by Golgi cell interneurons and under the control of long-term synaptic plasticity at the mossy fiber-granule cell synapse[12–14]. However, this evidence concerns elementary single neuron processes or, alternatively, neuronal ensembles[15–18] and is insufficient to explain how the granular layer microcircuit units would transform spatially distributed input patterns. A main hypothesis is that the cerebellum input stage operates as a spatially organized adaptive filter[19–21], implying that plasticity should be able to modify the processing of input patterns in the local microcircuit. Addressing this question requires simultaneous recordings of activity from multiple cerebellar neurons when a bundle of mossy fibers is activated. Under these conditions, short stimulus bursts in mossy fibers can be used to imitate those naturally occurring during punctuate facial stimulation[22,23] and generate response bursts in granule cells. Optical fluorescence calcium imaging techniques have recently been used to monitor multiple neuronal activities[24,25]. Here, we used a two-photon confocal microscope equipped with spatial light modulation (SLM-2PM) that proved capable of recording activity from several granular layer neurons simultaneously and was stable enough to monitor long-term changes in synaptic transmission[26]. Therefore, in principle, SLM-2PM should be able to map spatially distributed granular layer activity with single-cell resolution before and after the induction of long-term synaptic plasticity induced by high-frequency mossy fiber stimuli[27–29]. A further critical step is to extract the single spike patterns of neurons, which cannot be directly measured at the time resolution of calcium imaging but can be inferred using realistic modeling techniques. In recent years, granular layer modeling has advanced enough to guarantee good predictability of local microcircuit dynamics[30–33]. Finally, model predictions could be validated using voltage-sensitive dye imaging[15–18] to integrate multiple single neuron contributions. Techniques are therefore mature enough, in principle, to evaluate the cellular determinates of granular layer signal processing.

In this work, the combination of SLM-2PM and microcircuit modeling, followed by validation using VSD recordings, shows how long-term synaptic plasticity, once distributed over functional multi-neuronal units, creates spatial filters capable of tuning the time delay and gain of spike retransmission at the cerebellum input stage. These results provide a plausible basis to explain the spatiotemporal recoding[9–11] and adaptive filtering[19–21] of input spike patterns anticipated by theory.

## Results

**Multi-neuron responses of the granular layer following mossy fiber stimulation.** Calcium imaging (with Fura-2 AM) was performed using the SLM-2PM[26] in order to gain insight into the cellular organization of granular layer responses to mossy fiber bundle stimulation in acute cerebellar slices. We recorded up to 200 granule cells simultaneously, many of which (normally <=100) responded with fluorescence changes following the delivery of short stimulus bursts (10 pulses at 50 Hz) (Fig. 1a–c). Recordings were carried out both on the sagittal and coronal planes to account for potential asymmetries in granular layer responses. The multi-neuron maps ($\Delta F/F_0$ of granule cell responses to mossy fiber stimulation) covered a surface that was irregularly rounded in both the sagittal and coronal planes and showed decreasing activity from the core to the periphery. After applying the 10 μM of the GABA-A receptor blocker gabazine, the responses increased, indicating that Golgi cell synaptic inhibition limited the extension and intensity of granule cell activation.

The comparison of responses before and after GABA-A receptor blockade allowed us to calculate the excitatory-inhibitory (E–I) balance (Fig. 1b). The multi-neuron maps revealed evident core excitation (E–I > 0) and widespread peripheral inhibition (E–I < 0), confirming initial observations[26]. This arrangement was evident both in the sagittal and coronal planes.

Since multi-neuron maps were rather irregular, the spatial organization of neuronal activity was reconstructed by generating cumulative response maps from several recordings (5 sagittal slices and 4 coronal slices) (see "Methods" section and Supplementary Fig. s1) (Fig. 1d). These cumulative maps resembled the center/surround (C/S) organization observed using local field potential recordings and voltage-sensitive dye imaging[15,18]. The global activities (summing up the contribution of all active cells) in regions with either E–I > 0 or E–I < 0 were similar when comparing sagittal and coronal slices ($p = 0.8$ and $p = 0.3$, respectively; unpaired $t$-test; Fig. 1d). All subsequent experiments were carried out on sagittal slices.

**Spatial organization of long-term synaptic plasticity in the granular layer.** Since SLM-2PM recordings proved stable over time[26], they were used to investigate the spatial organization of plastic changes induced by high-frequency stimulation (HFS) of the mossy fiber bundle. Fura-2 signals were monitored in sagittal slices over a recording time of 60 min, reproducing a protocol similar to that used successfully in whole-cell recordings from granule cells[27–29,34] (Fig. 2a).

In the first set of experiments ($n = 7$ slices), SLM-2PM recordings were carried out with inhibition intact (control condition). HFS induced rapid and persistent changes in $\Delta F/F_0$ of granule cell responses configuring either calcium-related potentiation (CaR-P) ($99.8 \pm 6.5\%$, $n = 26$ cells; $p < 0.01$ ($p = 2.37e^{-7}$, Cohen's $d = 2.05$), unpaired $t$-test) or calcium-related depression (CaR-D) ($-35.1 \pm 3.9\%$, $n = 67$ cells; $p < 0.01$ ($p = 3.54e^{-5}$, Cohen's $d = 0.33$), unpaired $t$-test). Some granule cells showed changes in $\Delta F/F_0 <= \pm 20\%$ ($-3.7 \pm 1.6\%$, $n = 47$ cells; $p = 0.1$, N.S.) that were within the range of spontaneous response variability (cf. Fig. 2 in ref. [26]) and were not counted as either CaR-P or CaR-D.

In a second set of experiments ($n = 5$ slices), recordings were carried out with inhibition blocked by 10 μM gabazine. As in the control experiment, HFS induced rapid and persistent changes in $\Delta F/F_0$ of granule cell responses configuring either CaR-P ($86.8 \pm 4.7\%$, $n = 216$ cells; $p < 0.01$ ($p = 3.98e^{-8}$, Cohen's $d = 6.28$), unpaired $t$-test) or CaR-D ($-48.6 \pm 2.3\%$, $n = 51$ cells; $p < 0.01$ ($p = 4.72e^{-9}$, Cohen's $d = 0.24$), unpaired $t$-test). The cells that

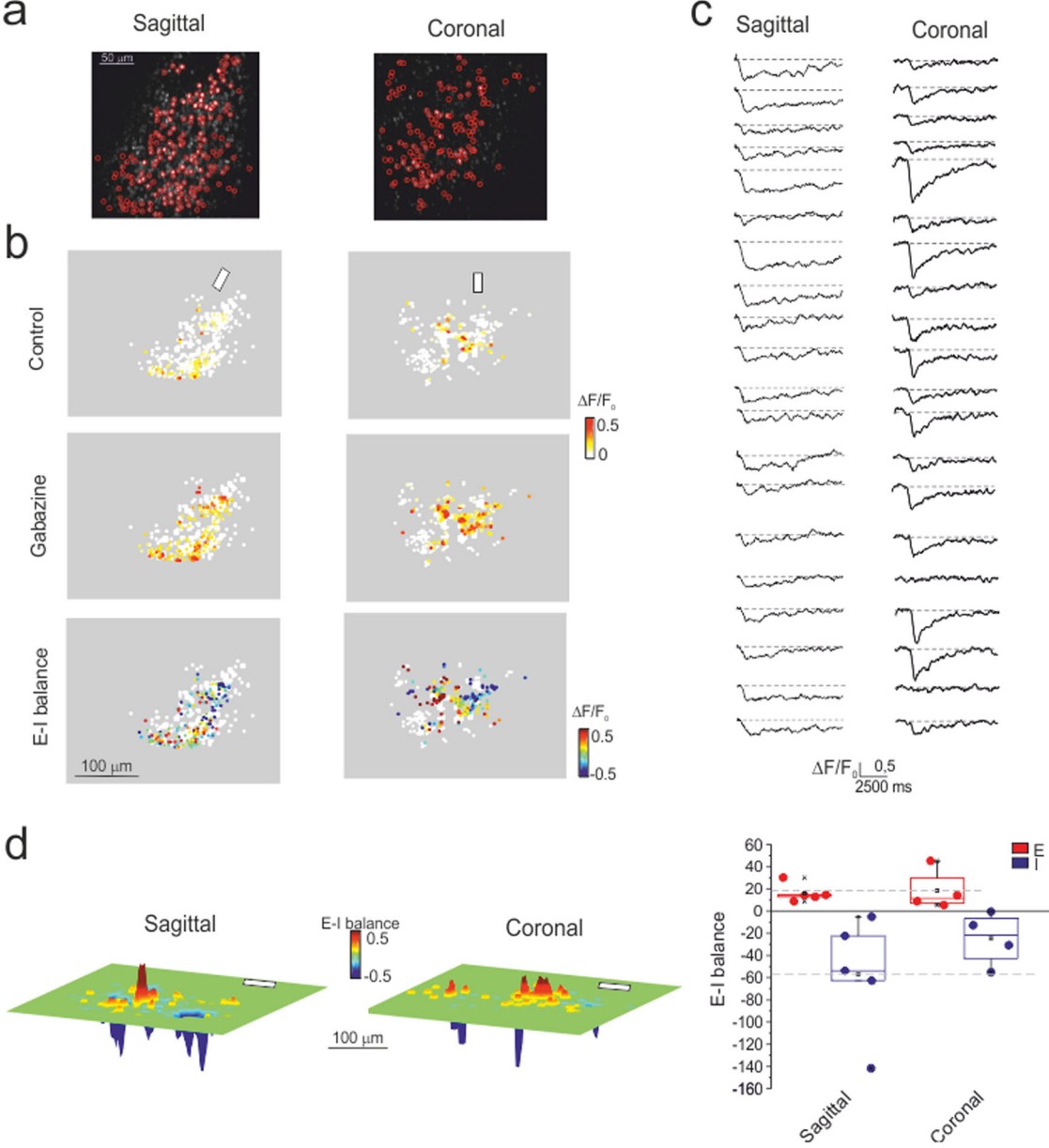

**Fig. 1 Multi-neuron maps of granule cell activity. a** SLM-2PM images of sagittal and coronal cerebellar slices (×20 objective, Fura-2 AM bulk loading). Some cells (red circles) were selected for calcium imaging. **b** Activity maps, taken from the cells selected in **a**, show the peak intensity ($\Delta F/F_0$) of the granule cell responses to mossy fiber stimulation (10 pulses at 50 Hz, white bars) on a color scale. Each dot corresponds to a neuron. In both sagittal and coronal slices, the maps show variable intensity of responses from cell to cell and a remarkable increase of activity after applying gabazine (10 μM, continuous bath perfusion). The $E$–$I$ balance maps (see "Methods" section) show that excitation or inhibition tend to prevail in distinct areas. **c** Examples of stimulus-induced $Ca^{2+}$ signals simultaneously detected from the somas of different granule cells. Fura-2 AM fluorescence signals appear as $\Delta F/F_0$ reductions induced by an intracellular calcium increase (6 points adjacent-averaging smoothing). **d** Cumulative $E$–$I$ balance maps show the spatial profile of granule cell activity ($n = 5$ independent sagittal slices and $n = 4$ independent coronal slices oriented with respect to the mossy fiber bundle, white bars). It should be noted that the most excited areas reside in the core and are flanked by inhibition. The box and whisker plots show that the cumulative activations of the E and I areas are similar in sagittal and coronal slices ($p = 0.8$ and $p = 0.3$, respectively; unpaired $t$-test).

showed changes in $\Delta F/F_0 < = \pm 20\%$ ($-3.1 \pm 1.7\%$, $n = 108$ cells; $p = 0.1$, N.S.) were not counted as either CaR-P or CaR-D.

To visualize the spatial organization of long-term response changes, cumulative plasticity maps were computed from several experiments (Fig. 2b; also, see "Methods" section). When synaptic inhibition was intact, the map showed a sharp CaR-P area surrounded by areas exhibiting CaR-D. When synaptic inhibition was blocked, CaR-P and CaR-D areas largely overlapped.

In synthesis, CaR-P and CaR-D were expressed both in the presence and absence of inhibition. The application of gabazine

markedly increased the number of cells showing CaR-P rather than CaR-D and therefore inverted the CaR-P/CaR-D ratio (Fig. 2c). The spatial organization of Fig. 2b reflected in the radial profiles of the number of cells showing CaR-P and CaR-D (Fig. 2d), which clearly demonstrated the inversion of the CaR-P/CaR-D ratio along with the extension of CaR-P beyond CaR-D when inhibition was blocked.

**Simulation of multi-neuron responses in a model of the cerebellar granular layer microcircuit.** Although SLM-2PM calcium

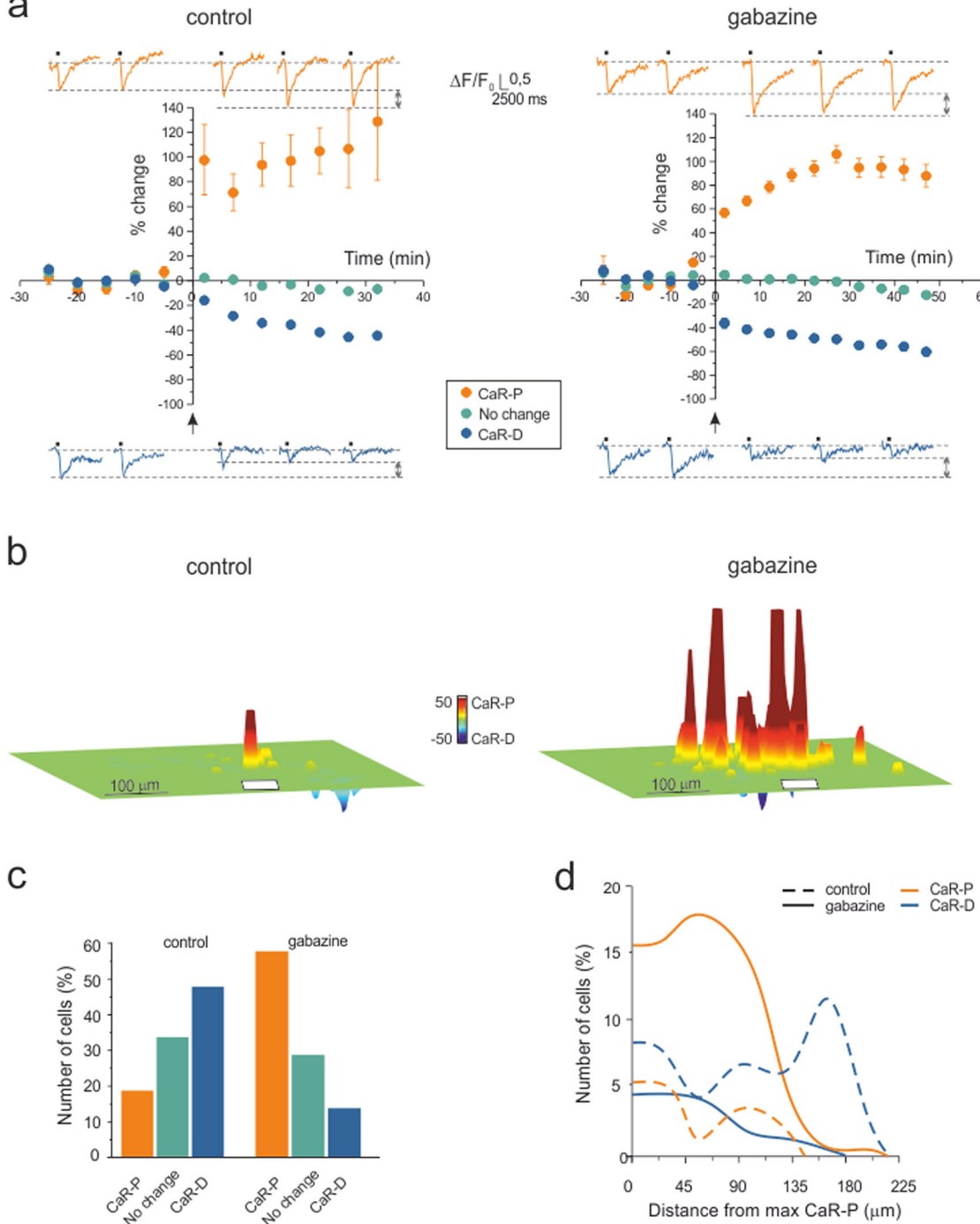

**Fig. 2 Long-term synaptic plasticity in multi-neuron recordings. a** Time course of granule cell calcium signal amplitude before and after plasticity induction (HFS was delivered to the mossy fiber bundle at the arrow) in the control condition and during gabazine perfusion. Persistent granule cell response variations appear as CaR-P (>20%) or CaR-D (<−20%) in both experimental sets. The changes <±20% are shown separately. The data points are the average of all the cells falling in the same category taken from available experiments (control: $n = 7$ independent slices, 140 cells; gabazine: $n = 5$ independent slices, 375 cells; mean ± SEM; some standard errors are small and fall inside the dots). The example traces show $Ca^{2+}$ signals before and after plasticity induction. **b** The cumulative plasticity maps show the different spatial organization of plasticity in the two experimental conditions. In the control condition, a sharp CaR-P area emerged surrounded by CaR-D regions (7 slices oriented with respect to the mossy fiber bundle, white bar), whereas during gabazine perfusion, a wide CaR-P area emerged, flanked by CaR-D areas (5 slices oriented with respect to the mossy fiber bundle, white bar). **c** Bar graphs showing the percentage of cells in each category shown in **a**. **d** Radial profiles of CaR-P and CaR-D in the two experimental sets in **a**–**c**. Note that when inhibition is blocked, CaR-P prevails over CaR-D by cell number and extension.

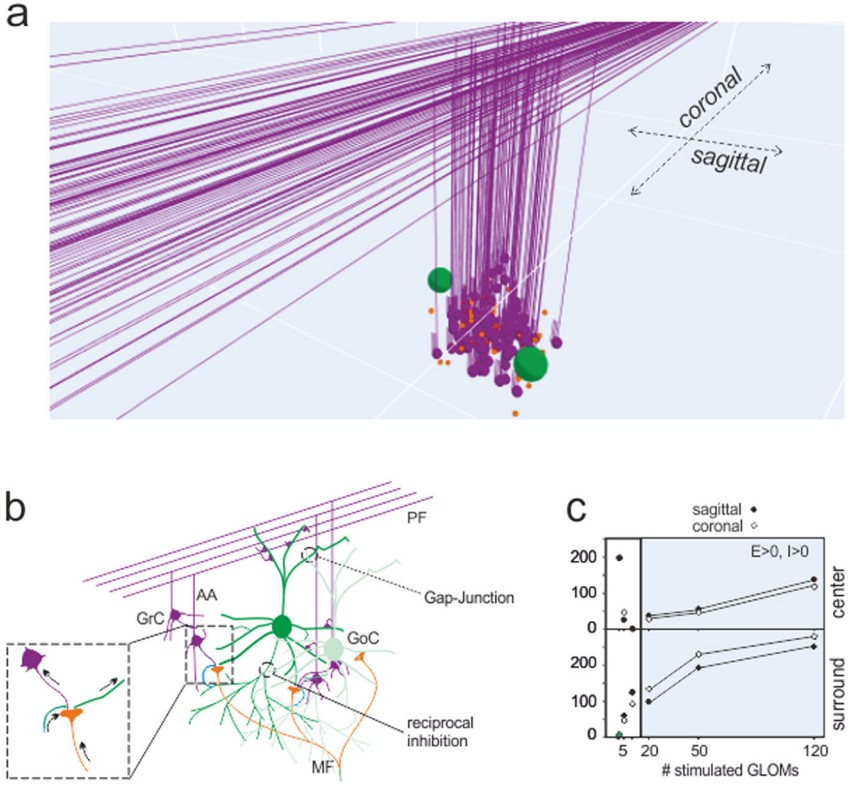

**Fig. 3 The granular layer network model. a** Scheme of the granular layer network and circuit wiring of a unit reconstructed in the model. The granule cells are activated by mossy fibers terminating in the glomeruli and are inhibited by Golgi cells. The Golgi cells are excited by mossy fibers, ascending axons and parallel fibers; are coupled by gap junctions and are connected by reciprocal inhibitory synapses. Network activity was elicited by stimulating the glomeruli. The full network reproduces an $800 \times 800 \times 150 \, \mu m^3$ portion of the granular layer, comprising 384000 granule cells, 914 Golgi cells and 29415 glomeruli. **b** Scheme of the granular layer wiring between granule cells, Golgi cells and glomeruli. Arrows indicate the flux of signals in and out of the glomerulus. **c** The model is used to determine the size of the microcircuit units generating responses compatible with SLM-2PM recordings such as those reported in Fig. 1. A C/S response structure emerged by activating more than 10 neighboring glomeruli. In this and the following figures, simulations were carried out by stimulating 50 adjacent glomeruli.

imaging provided a unique description of multicellular activity and plastic changes organized in space, the data were collected on a plane, and single neuron spikes could not be resolved. To predict the 3D structure of plasticity with spike-time resolution, we used a detailed data-driven model of the cerebellum granular layer (Fig. 3a–b; also, see "Methods" section and Supplementary Fig. s2).

The model was constructed using realistic connectivity and was equipped with detailed neuronal and synaptic properties[13,32,35]. The model was first used to determine the microcircuit unit generating responses compatible with experimental recordings in vitro using bursts delivered to the mossy fiber bundle (10 pulses at 50 Hz). Similar burst patterns were shown to occur during punctuation sensory stimulation in vivo[17,22,23,36–38] (Fig. 3b). Simulations were run by activating an increasing number of neighboring glomeruli (see "Methods" section). A C/S structure clearly emerged with >10 glomeruli, and the size of the structure increased with the number of glomeruli (Fig. 3c).

Under the assumption that there is a direct relationship between $\Delta F/F_0$, membrane depolarization and the number of spikes (see Fig. 4 in[26]), we compared SLM-2PM recordings to model output. According to simulations, a close match with the SLM-2PM imaging data of Fig. 1 was obtained when approximately 50 contiguous glomeruli were activated. The simulations with 50 glomeruli revealed the spatial distribution of spiking activity in the multi-neuron unit (Fig. 4a). Interestingly, in the inhibition ON condition, granule cells showed a variety of firing patterns, with different first-spike delays and numbers of spikes,

which were clearly correlated with their position in the unit (Fig. 4b). Conversely, in the inhibition OFF condition, the discharge pattern became more stereotyped, with short delays and high discharge frequencies. The maps obtained by integrating firing activity over the 3D model volume were consistent with those obtained experimentally (cf. Fig. 4c to Fig. 1d). The global activity of regions with either $E–I > 0$ or $E–I < 0$ was comparable in sagittal and coronal slices ($p = 0.75$ and $p = 0.57$, respectively; unpaired $t$-test), ruling out asymmetry in simulated, as well as in experimental C/S structures (Fig. 4c). All subsequent analysis of model simulations was carried out on the whole 3D response unit (also, see Supplementary Figs. s2–s3, and Supplementary Movies).

**Simulation of plastic changes.** One unresolved issue in experimental studies is how synaptic plasticity (long-term potentiation and long-term depression, LTP and LTD) is spatially distributed throughout the microcircuits. The model was therefore used to predict the expression of plasticity based on its cellular mechanisms. In granule cells receiving mossy fiber bursts, the membrane potential change is known to bring about corresponding modifications in $Ca^{2+}$ influx through NMDA receptor channels, which effectively trigger plasticity induction at mossy fiber-granule cell synapses[27,39]. LTP and LTD were simulated by measuring the average membrane potential of each granule cell during HFS, calculating neurotransmission changes using the Bear–Lisman rule[40–42], and then modifying mossy fiber release probability accordingly. This last step accounts for the release

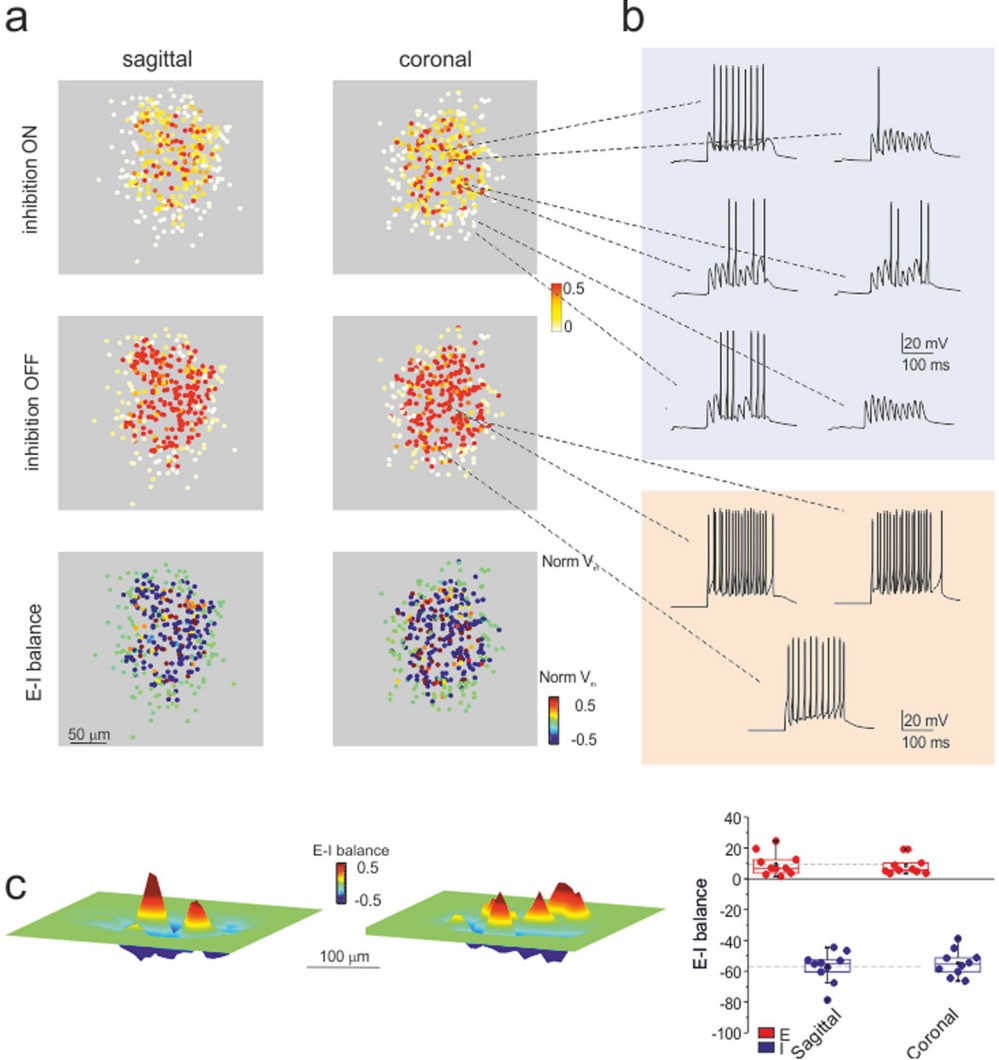

**Fig. 4 Simulated microcircuit response units. a** Sagittal and coronal sections of the response unit show the normalized membrane potential ($V_m$) of neurons in response to the activation of 50 contiguous glomeruli (10 pulses at 50 Hz). Each dot corresponds to a granule cell of the unit. In both sagittal and coronal sections, the maps show variable intensity of responses from cell to cell and a remarkable increase of activity after turning inhibition OFF. The E–I balance maps show distinct areas of prevailing excitation or inhibition. **b** Examples of simulated voltage traces related to granule cells from different positions in the microcircuit unit. Note that when inhibition is ON, granule cells show a variety of firing patterns; when inhibition is OFF, the pattern is more stereotyped. **c** E–I balance maps corresponding to simulations in **a**. The box-and-whisker plots show that activation of the E and I areas is similar in sagittal and coronal sections of the 3D activity maps ($n = 10$ independent simulations of sagittal sections and $n = 10$ independent simulations of coronal sections; $p = 0.75$ and $p = 0.57$, respectively; unpaired $t$-test).

probability values estimated by whole-cell recordings and quantal theory calculations[27,43] (Fig. 5).

The experiments reported in Fig. 2 were simulated with either inhibition ON or OFF (Fig. 6a and Supplementary Movies). Similar to experiments, LTP was dominant in the core and LTD was dominant in the periphery, and when inhibition was turned OFF, the LTP/LTD ratio increased and the core area extended towards the edge of the unit. These results reflect the arrangement of membrane voltage, NMDA current and $[Ca^{2+}]_i$ during induction (cf. Fig. 5). Since membrane depolarization was normally higher in the core than in the periphery, the probability of having LTP was also higher in the core, while the probability of having LTD was higher in the periphery (Fig. 6a).

A specific advantage of simulations was to obtain insight into the firing properties of the multi-neuron units (Fig. 6b). After plasticity, the number of firing granule cells increased, and they responded with a shorter delay, with more spikes and at a higher frequency (see Supplementary Figs. s2, s3, and s4). The changes

in gain and lag induced by plasticity were more accentuated in the core compared to the periphery of the response unit, reflecting the concentration of LTP in the core and of LTD in the periphery. Thus, synaptic plasticity eventually increased the contrast between these two regions of the response unit, with the effect of focusing the transmission channel.

Example traces are shown in Fig. 6c along with the underlying synaptic mechanisms. Previous experimental observations using VSD recordings revealed that GABA receptor-mediated inhibition and NMDA receptor-mediated depolarization are key to explaining gain changes in the granular layer[16,30]. Interestingly, the model predicted that gain reflected the balance between the activation of NMDA-AMPA and GABA-A currents. This balance was higher in the core than in the periphery of the unit (Fig. 6d) due to the radial decrease of active mossy fiber-granule cell dendrites against maintained inhibition through the extended Golgi cell axonal plexus (also, see Supplementary Fig. s2). LTP enhanced neurotransmitter release along with NMDA and

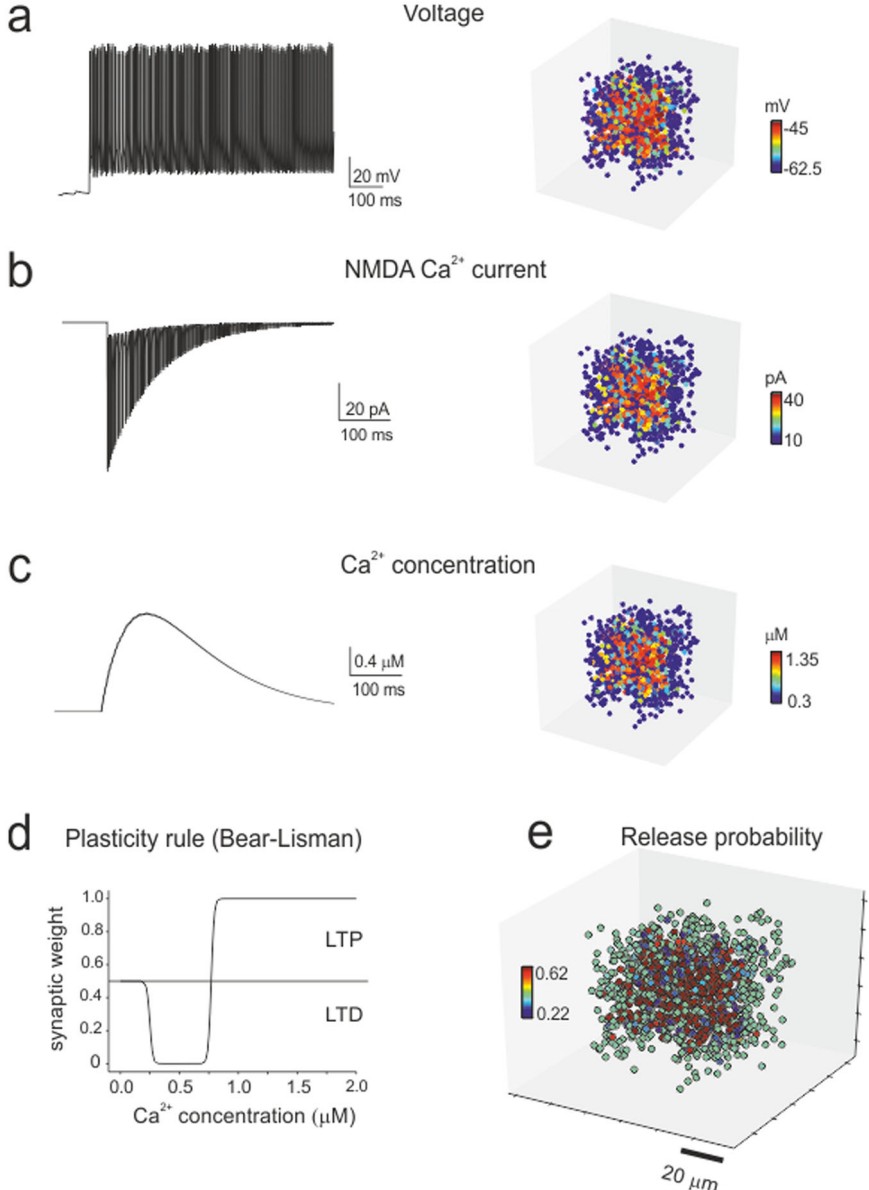

**Fig. 5 Simulation of synaptic plasticity. a–c** Cellular firing caused by HFS (the same as in the experimental induction protocol) determines $Ca^{2+}$ influx through synaptic NMDA receptor channels and the consequent increase in intracellular $Ca^{2+}$ concentration. The membrane voltage (average voltage), NMDA current (integral of the current) and $Ca^{2+}$ concentration (peak concentration) vary from cell to cell depending on the local connectivity inside the microcircuit unit, as shown in the 3D activity maps. **d** The Bear–Lisman rule was used to calculate the neurotransmission changes induced by the $Ca^{2+}$ concentrations reached during HFS. **e** The mossy fiber release probability changed accordingly (plotted here over the receiving granule cells) to account for LTP or LTD expression.

AMPA receptor-mediated currents, which concurred in raising granule cell synaptic excitation. The GABA-A receptor-mediated current also increased, reflecting enhanced activation of the granule cell-Golgi cell loop through the ascending axon synapse (also, see Supplementary Fig. s5).

**Model prediction and experimental validation of spatio-temporal recoding of mossy fiber inputs**. The granular layer has been proposed as a location in which spatiotemporal recoding of incoming mossy fiber inputs can be performed[9], but the cellular basis of this activity has remained unclear. Here, we have used the model to analyze how synaptic plasticity can regulate the gain[12,13] of the input-output function. The gain was assessed by applying input bursts (5 spikes) at different frequencies (10–500 Hz) to

explore the physiological bandwidth[16] in the control condition and after the induction of synaptic plasticity.

Granule cells showed frequency-dependent responses characterized by gain plots that were fitted with logistic curves yielding gain and cutoff frequency estimates (Fig. 7a). Granule cells were divided into groups with an area of $2.5 \times 2.5\ \mu m^2$ pixels for each group, where their voltage responses were summed and averaged over 1-ms time windows. This analysis showed that different pixels had different gain curves that were grouped per cutoff frequency range (10–20 Hz, 20–50 Hz, 50–100 Hz, 100–200 Hz, and 200–500 Hz). These properties were spatially distributed, with higher gain and lower cutoff in the center of the response unit (Fig. 7b).

After the induction of plasticity, the gain curves maintained similar shapes, but the maps changed. The gain increased in the

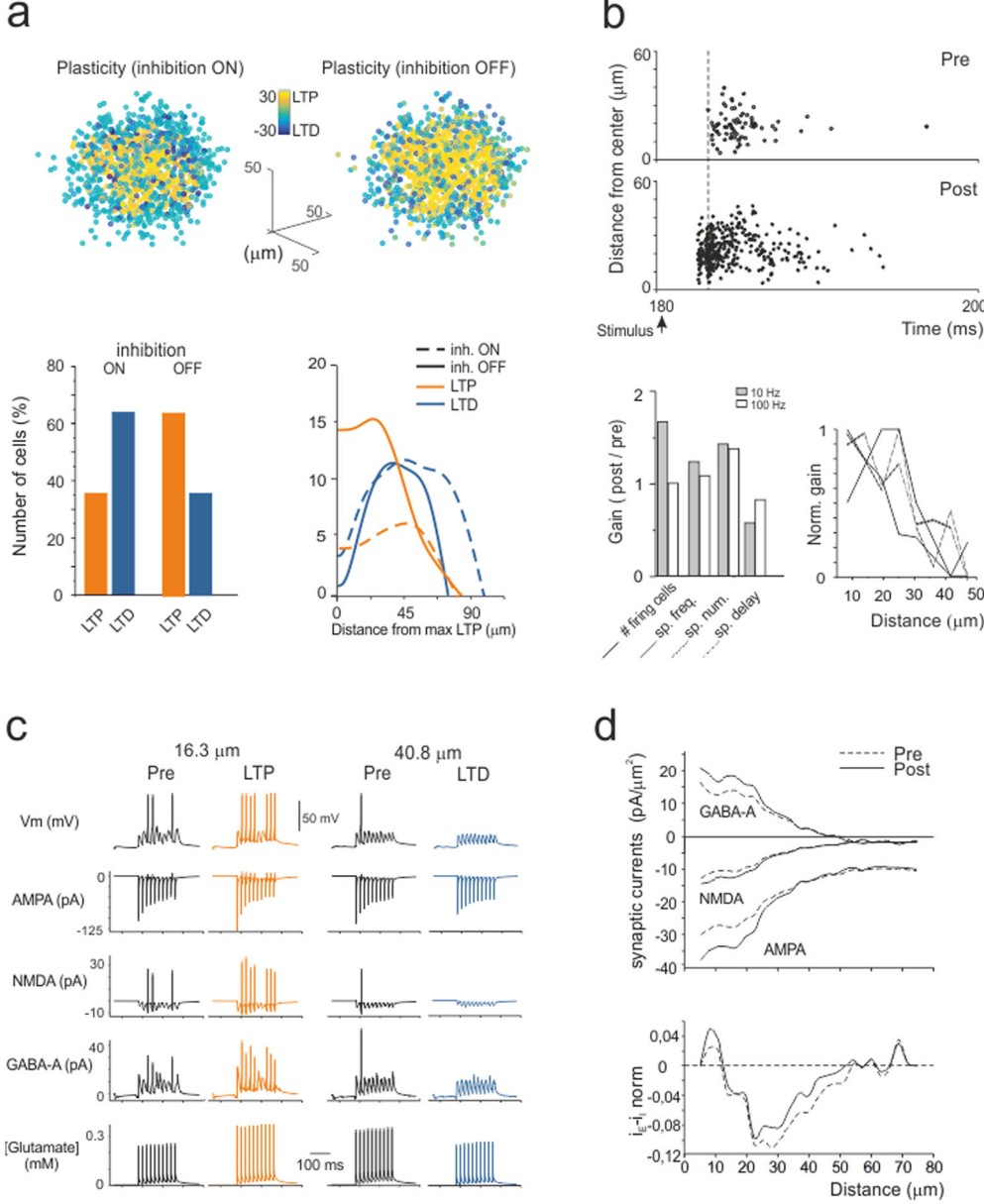

**Fig. 6 Simulation of plastic changes. a** 3D maps of simulated plastic changes with inhibition either ON or OFF. The bar graph shows the percentage of cells generating either LTP or LTD. The plot shows the radial profiles of LTP and LTD in both conditions. Note that when inhibition is blocked, LTP prevails over LTD both in the number of granule cells and in extension towards the edge of the recorded area, mirroring experimental observations (cf. Fig. 2). **b** This panel illustrates the main spike changes following the induction of synaptic plasticity. The raster plot shows the response of all granule cells in the unit to an input burst (5 pulses at 50 Hz), both before and after induction. Note that after induction, the cells respond more rapidly, with more spikes and at higher frequency. This visual impression is quantified in the bar graphs that show the gain (post/pre) in the number of firing cells, spike frequency, spike number and first-spike delay (5 pulses at 10 Hz and at 100 Hz). A full description of the frequency dependence of these parameters is given in Supplementary Fig. s4. The plot shows the normalized gain of these parameters as a function of distance from the center of the unit (for ease of comparison, the change in delay has been inverted). Note that the tuning of the spike frequency and first-spike delay decreases from the core to the periphery of the unit. **c** This panel illustrates the main synaptic changes following the induction of synaptic plasticity in two example granule cells responding to test stimuli (10 pulses at 50 Hz), one showing LTP in the core and the other exhibiting LTD in the periphery of the unit (16.3 μm and 40.8 μm, respectively; inhibition ON). Below membrane voltage, the traces show simulated AMPA, NMDA, GABA-A receptor-mediated ionic currents and glutamate concentration. Note that LTP is associated with potentiated and anticipated spike discharge along with an increase in glutamate concentration and AMPA and NMDA currents, whereas the opposite occurs with LTD. **d** This panel illustrates the spatial distribution of synaptic currents (AMPA, NMDA, and GABA-A) in response to test stimuli (10 pulses at 50 Hz) before and after the induction of plasticity (inhibition ON). The space from the core of the unit was measured and divided into 5 μm concentric spherical crowns. Note that synaptic current density changes after plasticity are more marked in the core of the unit. The current density changes were normalized to show the balance between excitatory and inhibitory currents before and after plasticity. The balance is such that excitatory currents prevail in the core and inhibitory currents prevail in the periphery of the unit.

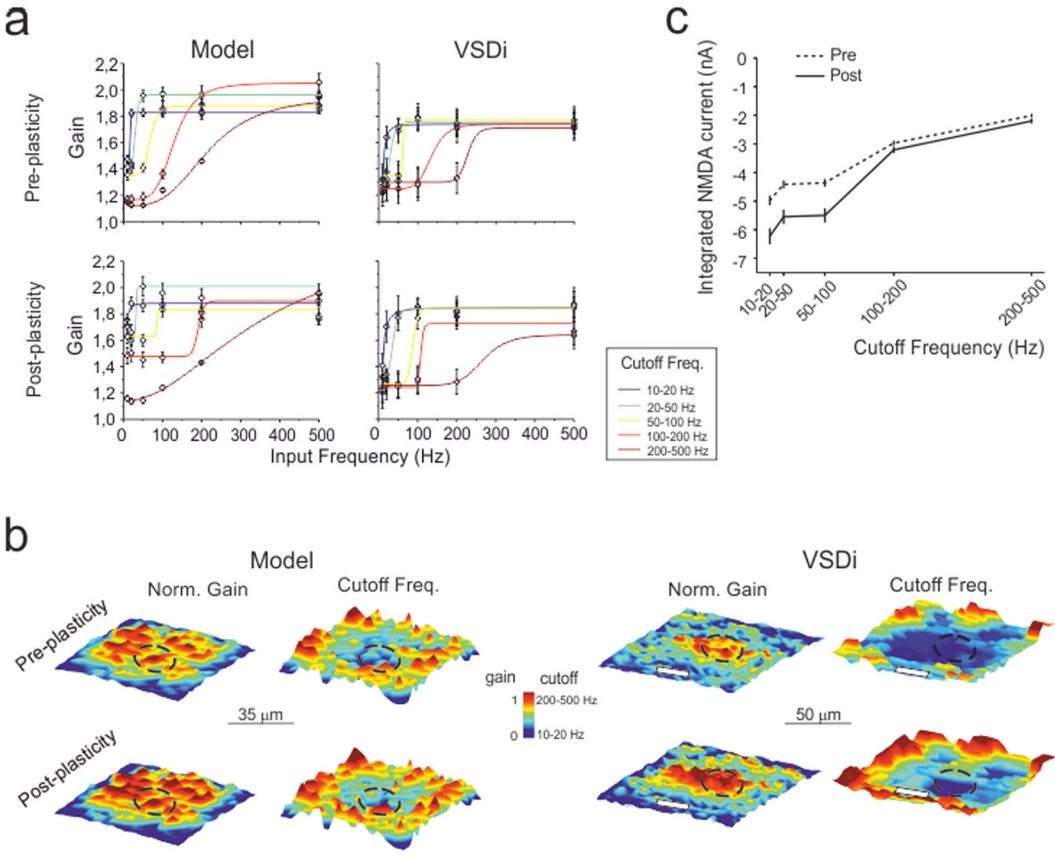

**Fig. 7 Filtering properties of the granular layer: simulation and experimental validation. a** The plots show granule cell output gain as a function of input frequencies before and after plasticity, both from model simulations ($n = 5$ independent slice simulations) and VSDi experiments ($n = 5$ independent slices). The curves show average sigmoidal fittings for pixels showing similar filtering properties. **b** Average gain and cutoff maps were obtained by centering pixels showing maximal response and aligning maps along the mossy fiber bundle (white bars). After plasticity, the gain and cutoff frequency show a marked increase in the core. Note the remarkable similarity between the simulation and VSDi results. The dotted circles indicate the area used to compute core parameter changes. **c** The NMDA current density with respect to the cutoff frequency in the simulated granule cell clusters. Note the negative correlation (more current at low frequency) and the NMDA current increase after the induction of plasticity. Error bars represent the standard error of the mean (SEM).

center of the response unit along with an increase in cutoff frequency. Therefore, the model predicted that different parts of the response unit had different frequency-dependent transmission properties, which were changed by long-term synaptic plasticity. These observations show spatiotemporal recoding of incoming mossy fiber patterns. According to the analysis shown in Fig. 6c-d, the ability to transmit at specific cutoff frequencies was correlated with the activation of NMDA receptor-mediated currents in the corresponding neuronal clusters (Fig. 7c).

This simulation was purposefully designed to be comparable with a validation experiment using voltage-sensitive dye imaging (VSDi)[16,44]. VSD recordings were performed in parasagittal slices using mossy fiber stimuli like those used in simulations (Fig. 7a; cf. Fig. 7b), and the signals were sampled by a $2.5 \times 2.5 \, \mu m^2$ pixelated CMOS sensor at 1 kHz. Since the VSDi signal reflects the sum of all membrane potential changes occurring in a pixel averaged over the recording time, the VSDi recordings can be directly compared to simulations. VSDi signal patterns closely matched modeling predictions. The average change [(post-pre)/pre] in the core was as follows: gain VSD + 2.9 ± 0.3% vs. gain model +2.9 ± 0.4% ($n = 50$, $p = 0.95$; unpaired $t$-test), cutoff VSD + 36.7 ± .8% vs. cutoff model +27.1 ± 14.5% ($n = 50$, $p = 0.52$; unpaired $t$-test).

**Simulation of filtering channels tuned by synaptic plasticity.** A critical prediction of the granular layer circuit is its ability to

perform spatial pattern separation[9]. Here, this property should emerge when activating partially overlapping granular layer fields through mossy fiber bundles sharing common elements (Fig. 8a). Indeed, the injection of different frequency patterns in two partially overlapping bundles resulted in distinct outputs reflecting the frequencies of the two inputs (Fig. 8b). Therefore, the network was able to spatially separate output patterns despite their partial overlapping input.

The filtering capability of the network was further simulated by comparing three units located in different positions (Fig. 8a). The hypothesis is that by virtue of the variable wiring with mossy fibers and Golgi cells, the granule cells in these units are in a different initial state and will thus develop different membrane potential changes and plasticity (e.g., see ref. [21]). We have therefore stimulated three units with input spike patterns containing mixed frequency components (e.g., in the theta, alpha, beta and gamma bands). The spectral patterns of the firing frequencies transmitted by the three units before and after plasticity are shown in Fig. 8c. Clearly, the three units are not identical, and after plasticity, unit *I* becomes the best in transmitting at ~50 Hz and ~80 Hz, unit *II* best transmits at ~10 Hz, and unit *III* best transmits at ~70 Hz. Thus, the three units clearly differentiate their transmission properties after plasticity and behave as differential filters.

We then considered how the transmission properties were distributed into the response unit by performing a fast Fourier

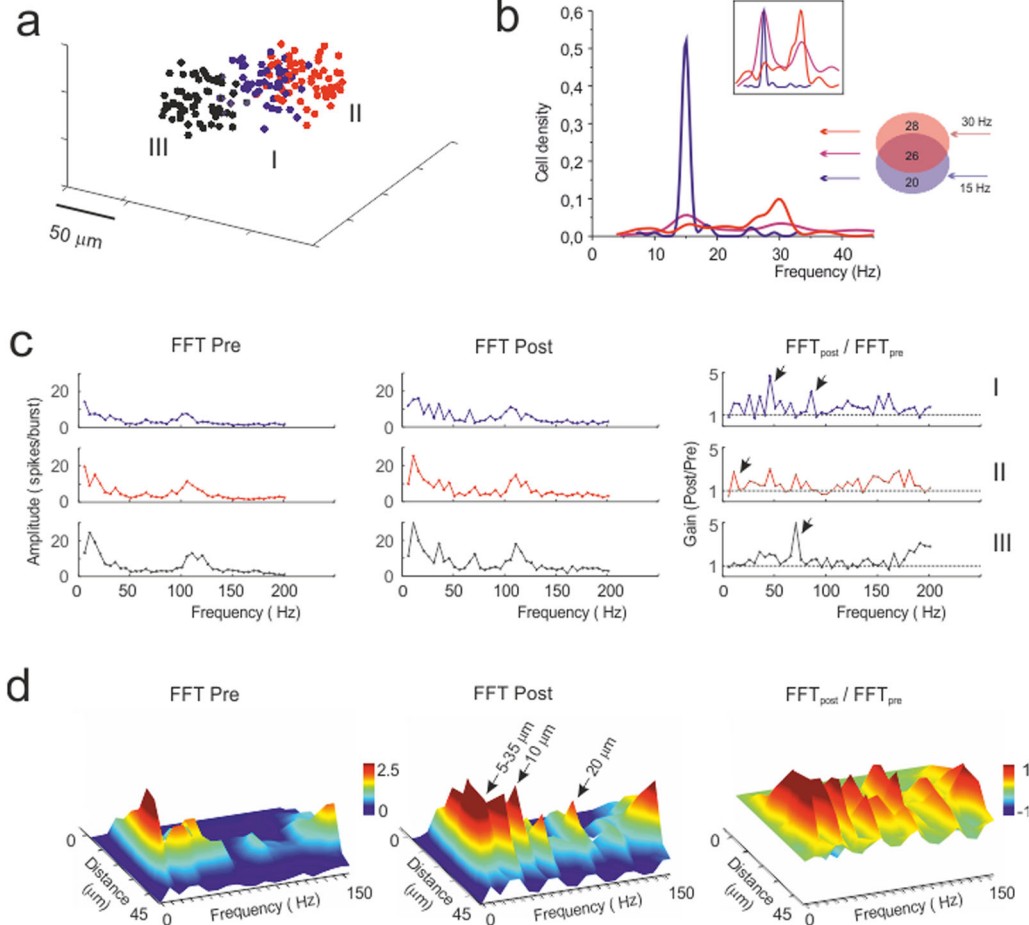

**Fig. 8 The basis of differential filtering in granular layer response units. a** 3D localization of the glomeruli belonging to 3 different mossy fiber bundles and activating 3 different units. **b** The plot shows the frequency discrimination operated by granule cells belonging to two of the units in **a**, which have partially overlapping input (26 out of 74 glomeruli are shared between the two bundles). The units are stimulated with different frequency patterns (unit I at 15 Hz; unit II at 30 Hz). Note that granule cells emit spikes over distinct frequency bands corresponding to the inputs in the overlapping region. **c** The units shown in **a** are stimulated alternatively with a mossy fiber pattern containing mixed frequencies (25% at 20 Hz, 25% at 30 Hz, 50% at 4 Hz with superimposed 100 Hz bursts). The plots show the fast Fourier transform (FFT) of the 3 units (all output spikes pooled together) before and after plasticity. The ratio (FFT$_{post}$ /FFT$_{pre}$) for each unit shows the emergence of preferred frequencies at which the units better transmit spikes; e.g., unit #1 at 50 Hz and 80 Hz, unit #2 at 10 Hz, and unit #3 at 70 Hz (arrows). **d** The plots show the FFT of unit #1 after separating the output spikes depending on their distance from the center of the unit before and after plasticity. The ratio (FFT$_{post}$/FFT$_{pre}$) shows the regions that change their transmission properties for each frequency. Transmission generally improves after plasticity with the 40 Hz peak located 10 μm from the center of the response unit and the 80 Hz peak located 20 μm from the center of the response unit.

transform in concentric spherical crowns (Fig. 8d). Notably, frequencies were differentially filtered at specific distances from the center of the unit and, after plasticity, a generalized enhancement of transmission was observed across the whole unit.

## Discussion
The spatial filtering properties of the cerebellar granular layer revealed in this work support a critical prediction of motor learning theory[7,9,10,21]. The theory suggests that input signals entering the cerebellum undergo a process of combinatorial expansion recoding, in which the signals are separated into different components and transformed before being conveyed to the molecular layer. Combinatorial expansion was thought to originate from the divergence/convergence ratio of mossy fiber-granule cell synapses (introducing the concept of spatial pattern separation) and recoding was thought to originate from the inhibitory control that Golgi cells exert on granule cell activity. Then, adaptation needed for learning was thought to occur

through plasticity at parallel fiber-Purkinje cell synapses. This view has been implemented into the adaptive filter model[19–21], in which information is split through different channels processing granular layer input signals. Our observations show that such channels can indeed be generated through the contribution of granule and Golgi cells aggregated into multi-neuronal units. Interestingly, these channels proved themselves tunable by synaptic plasticity, adding a further dimension to cerebellar granular layer filtering properties.

When a mossy fiber bundle was activated by input bursts imitating those reported in vivo following punctuate sensory stimulation[22,23], granular layer responses were organized in units similar to those observed using multielectrode arrays[15], voltage-sensitive dye imaging[18] and two-photon microscopy[26]. The units showed an irregular rounded shape in both sagittal and coronal sections. The core of the units was more excitable than the periphery, and excitation became more prominent and extended when inhibition was blocked pharmacologically. The units underwent remodeling following high-frequency mossy fiber

stimulation, which made the difference between the core and periphery more distinct. The underlying cellular changes were CaR-P and CaR-D and had magnitude and time courses resembling those of LTP and LTD reported at the mossy fiber-granule cell synapse[27–29,34]. Interestingly, approximately 20% of the granule cells underwent CaR-P and 50% underwent CaR-D, but this proportion was almost inverted when synaptic inhibition was blocked by gabazine. Moreover, whereas CaR-P normally involved fewer cells than CaR-D and was confined to the internal part of the unit, in the absence of inhibition, the number of cells showing CaR-P surpassed the number of cells showing CaR-D and extended to the edge of the unit.

Model simulations at single-cell spike resolution showed that units, such as those observed with SLM-2PM, could be generated by activating approximately 50 adjacent glomeruli contacting ~300 granule cells. These units occupied an irregular rounded volume with the most excited cells in the core. Turning off Golgi cell inhibition changed the response in a manner that was consistent with the calcium imaging experiments using pharmacological inhibition (cf. Fig. 4 to Fig. 1). Model validation was given by VSDi recordings, which provided the spatiotemporal integral of membrane potential changes occurring in a given network volume[16,45]. The size and properties of the units resemble those obtained from the reconvolution of local-field potentials elicited by punctuate facial stimulation[17], which predicted 11% discharging granule cells at rest and an increase to approximately 30% during plasticity over approximately 250 granule cells. Therefore, the units obtained in vitro and in silico are likely to represent the counterpart of the dense clusters activated in the granular layer in vivo. Dense clusters may sustain stereotyped patterns of functional connectivity[46], and their activation has been revealed in the acquisition of predictive feedback signals during motor learning[47] and in the encoding of expectation reward[48].

High-frequency stimulation of mossy fibers is known to induce NMDA-receptor-dependent LTP and LTD in granule cells[27], which were previously shown to conform to the Lisman rule for $Ca^{2+}$-dependent induction[27,39]. The Bear–Lisman[40–42] plasticity model showed that the differential distribution of membrane potential, NMDA receptor activation and calcium influx in the granule cells of units could indeed account for the plastic changes obtained with SLM-2PM recordings. Simulations correctly predicted that LTP was more concentrated in the core and LTD was more concentrated in the periphery of the unit. Moreover, the LTP/LTD ratio was inverted, and LTP extended towards the edge of the unit when inhibition was switched off (further details are provided in the Supplementary Fig. s4). Therefore, LTP and LTD predicted by the Bear–Lisman model were sufficient to explain the spatial properties of experimental CaR-P and CaR-D. Additional mechanisms, such as non-synaptic plasticity[17,29] and the intervention of mGlu-Rs and $Ca^{2+}$ release from internal stores[27,39], as well as combinations of presynaptic and postsynaptic expression mechanisms[18,28], could contribute to changes in the LTP/LTD landscape in the unit. In these cases, more precise experimental information would be needed to implement the corresponding induction mechanisms and obtain reliable modeling predictions.

Model simulations predicted that granular layer units express filtering properties tuned by long-term synaptic plasticity. Since the changes in granule cell responses were more evident at low frequencies than at high frequencies and since LTP was condensed in the core and LTD was condensed in the surroundings, plasticity created a spatial low-pass filter channeling rapid high-gain high-contrast information through the unit core. In the core, raising the mossy fiber frequency from 10 Hz to 100 Hz almost doubled the number of firing granule cells and halved the first spike delay but only slightly increased the frequency and number of emitted spikes (cf. Fig. 6b), akin to a burst-burst transmission scheme[23].

The fine-grained properties of the units were peculiar, in that neuronal clusters dispersed inside the unit were able to process specific frequency bands. In addition to the model predicting these properties, a similar pattern emerged in VSDi recordings, showing that different neuronal clusters have preferential gain and cutoff frequency. This capacity also emerged in the presence of two partially overlapping bundles, providing evidence for spatial pattern separation[9].

The model revealed that differential activation of AMPA, NMDA, and GABA-A receptors[49] and local changes in glutamate release probability (which regulates short-term synaptic dynamics under LTP and LTD control[13]) explained cluster selectivity. In particular, NMDA receptors proved quite important not only for plasticity induction but also for regulating signal integration. The granule cell subtypes (adapting, non-adapting and accelerating) discovered recently may further bias single neuron filtering depending on the expression of TRPM4 channels[50], and membrane potential integration by different K channels may favor the emergence of resonance peaks in the theta band[30,51].

In summary, these simulations strongly support the concept that mossy fiber-granule cell plasticity is fundamental to tuning the filtering properties of the cerebellar granular layer. The mechanism through which plastic changes might evolve in life and achieve balance with the stability of local computation[52] introduce interesting issues that remain to be investigated.

In conclusion, microscopic analysis of granular layer transmission properties modulated by long-term synaptic plasticity supports the adaptive filter model of the cerebellum[19–21], although there are notable differences from the original beliefs. First, granular layer filtering involves spatially organized multicellular units including ~50 glomeruli and ~300 granule cells. Second, granular layer filtering is tuned by local mechanisms of long-term synaptic plasticity. Third, this plasticity involves both LTP and LTD in a spatially organized manner. These concepts support the hypothesis that the granular layer can form multiple signal filtering channels regulating spike delay and discharge gain[14]. These channels can differentiate the cerebellar cortex filtering properties[53,54], focus the signal in vertical beams traveling towards the molecular layer (see refs. [11,33,55]) and control signal integration and coincidence detection in Purkinje cells[56,57]. Thus, plasticity-controlled distribution of spike timing and discharge can remarkably impact mutual information transfer at the cerebellum input stage[58,59] and influence subsequent computations in the molecular layer and Purkinje cells.

## Methods

In this work, experiments and simulations were co-designed so that the experiments were analyzed trough a realistic model of the underlying microcircuit. Results from experiments and simulations were analyzed using similar routines to enhance a direct comparison.

**Experimental recordings**. All experimental protocols were conducted in accordance with international guidelines from the European Union Directive 2010/63/EU on the ethical use of animals and were approved by the ethical committee of Italian Ministry of Health (639.2017-PR; 7/2017-PR).

**Slice preparation and solution**. Cerebellar granular layer activity was recorded from the vermis central lobe of acute parasagittal or coronal cerebellar slices (230 μm thick) obtained from 18-day-old to 25-day-old Wistar rats of either sex. Slice preparation was performed as following: rats were decapitated after deep anesthesia with halothane (Sigma, St. Louis, MO), the cerebellum was gently removed and the vermis was isolated, fixed on a vibroslicer's stage (Leica VT1200S) with cyano-acrylic glue and immersed in cold (2–3 °C) oxygenated Krebs solution containing (mM): 120 NaCl, 2 KCl, 2 $CaCl_2$, 1.2 $MgSO_4$, 1.18 $KH_2PO_4$, 26 $NaHCO_3$, and 11 glucose, equilibrated with 95% $O_2$–5% $CO_2$ (pH 7.4). Slices were allowed to recover at room temperature for at least 40 min before staining[26,49,60].

**Multi-neuron two-photon calcium imaging**. Two-photon calcium imaging experiments were performed through a spatial light modulator two-photon microscope (SLM-2PM)[26,61]. The SLM-2PM uses a spatial light modulator device (SLM, X10468-07, Hamamatsu, Japan) to perform computer-driven holographic microscopy. This device allows to modulate the phase of a coherent laser-light source (Chameleon Ultra II, Coherent, USA) and to generate different spatial illumination patterns on the sample plane, by taking advantage of optical Fourier transform. Custom-developed routines written in Python 2.7 (PSF, 9450 SW Gemini Dr. Beaverton, OR 97008, USA) and based on the iterative-adaptive Gerchberg-Saxon algorithm were used to compute the phase distribution corresponding to a desired illumination pattern[62,63] (for more details see ref. [61]). In this way it is possible to split the laser beam into multiple beamlets that can be simultaneously directed onto different points of interest in the sample, thus allowing to record from multiple neuron simultaneously, while maintaining the single cell resolution.

**Two-photon data acquisition and analysis**. Experiments were performed at depths of ~50–60 μm, using a ×20 1.0 NA water-immersion objective (Zeiss Plan-APOCHROMAT) while the laser power was set ≤6 mW/laser beamlet. Cerebellar slices were bulk loaded with a 50 μg aliquot of Fura-2 AM (Molecular Probes, Eugene, OR, USA) previously dissolved in 48 μl of Dimethyl Sulfoxide (DMSO, Sigma Aldrich) and 2 μl Pluronic F-127 (Molecular Probes) and mixed with 2.5 ml of continuously oxygenated Krebs solution. The slices were placed in this solution and maintained at 35 °C for 40 min in the dark for bulk loading[64,65]. Then the slices were gently positioned in the recording chamber and immobilized using a nylon mesh attached to a platinum Ω-wire to improve tissue adhesion and mechanical stability. Oxygenated Krebs solution was perfused during the whole experiment (2 ml/min) and maintained at 32 °C with a Peltier feedback device (TC-324B, Warner Instruments, Hamden, CT). Two-photon images were acquired through a high-spatial resolution CCD camera (CoolSnap HQ, Photometrics, Tucson, USA) that covered a field of view of about 220 × 300 μm$^2$.

Following a validated procedure[26,61], granule cells were identified by evaluating the shape and size of the soma, thus differentiating them from the mossy fiber boutons and the Golgi cells. Granule cells activity was elicited by electrical stimulation of the mossy fiber bundle (10 pulses at 50 Hz repeated 3 times at 0.08 Hz to improve the signal-to-noise ratio, S/N) with a large-tip patch-pipette (~10–20 μm tip) filled with extracellular Krebs' solution, via a stimulus isolation unit. The stimulus-induced fluorescence calcium signals were acquired through a high-temporal resolution CMOS camera (MICAM Ultima, Scimedia, Japan), connected through an I/O interface (BrainVision, Scimedia, Japan) to a PC controlling illumination, stimulation and data acquisition. The final pixel size was 4.6 × 4.6 μm$^2$. Data were acquired at a frequency of 20 Hz and displayed by Brainvision software. Granule cells showed calcium signal variations peaking around 150 ms after the stimulus[26,61]. The stimulus-induced fluorescence signals were analyzed off-line by evaluating the peak amplitude fluorescence variations normalized to $F_0$, i.e., $(\Delta F(n)/F_0 = (F(n)-F_0)/F_0$, where $F_0$ was the mean resting fluorescence sampled for 1400 ms before triggering electrical stimulation and $F(n)$ was the fluorescence intensity at the $n$th frame. Given a peak amplitude of 0.2–0.9 and a noise standard error of 0.05–0.08, the $\Delta F/F_0$ S/N was about 8-fold, ensuring a reliable measurement of peak response amplitude.

In a set of experiments, the granule cells calcium responses were acquired before and after the perfusion of a GABA-A receptors inhibition blocker (10 μM gabazine, SR95531, Abcam). In another set of experiments the granule cells calcium responses were acquired before and after the delivery of a high-frequency mossy fibers stimulation protocol (HFS, 100 pulses at 100 Hz, known to induce long-term synaptic plasticity at the mossy fiber-granule cells synapses[27,29,39]), either in control (Krebs) or with inhibition blocked (Krebs + gabazine).

The neurons were considered to show long-term changes when their $\Delta F/F_0$ persistently changed by >± 20% after HFS.

As explained in refs. [15,18], the excitatory-inhibitory (E–I) balance maps were computed for each experiment as

$$E - I = (E_{norm} - I_{norm})/E_{norm} \qquad (1)$$

where $E_{norm}$ is the response intensity in control condition and $I_{norm}$ is the response intensity variation after gabazine perfusion (both normalized to their maximum response), so that E–I values ranged from 1 (maximal excitation) to −1 (maximal inhibition).

Cumulative response maps were computed by centering the individual maps on the cells showing maximum response in control and by aligning them along the mossy fiber bundle. Cumulative plasticity maps (i.e., those obtained by comparing responses amplitude before and after plasticity induction) were computed by centering the individual maps on the peak value (maximum potentiation) and by aligning them along the mossy fiber bundle. The cumulative maps were smoothed with a sliding box filter (3 × 3 pixels).

**Voltage sensitive dye imaging (VSDi) data acquisition and analysis**. VSDi experiments were performed as reported in previous papers[15,16,44]. Briefly, the slices were incubated for 30 min in oxygenated Krebs solution added with 3% Di-4-ANEPPS stock solution mixed with 50% fetal Bovine Serum (Molecular Probes).

The dye (Di-4-ANEPPS, Molecular Probes) was dissolved and stocked in Krebs with 50% ethanol (SIGMA) and 5% Cremophor EL (a Castor oil derivative, SIGMA). Perfusion of standard extracellular solution (2–3 ml/min) maintained at 32 °C with a feed-back temperature controller (Thermostat HC2, Multi Channel Systems, Reutlingen, Germany) was performed during the recording session. Mossy fibers were stimulated as explained for SLM-2PM experiments (5 pulses at 10–500 Hz repeated 10 times at 0.1 Hz to improve the signal-to-noise ratio, S/N). The recording chamber was installed on an upright epifluorescence microscope (BX51WI, Olympus, Europa Gmbh, Hamburg, Germany), equipped with a ×20 objective (XLUM Plan FL 0.95 NA)[66]. The light generated by a halogen lamp (150 W, MHF-G150LR, MORITEX Corp., Tokyo, Japan) was controlled by an electronic shutter (model0, Copal, Co., Tokyo, Japan) and then passed through an excitation filter (λ = 530 ± 10 nm), projected onto a dichroic mirror (λ = 565 nm) and reflected toward the objective lens to illuminate the specimen. Fluorescence generated by the tissue was transmitted through an absorption filter (λ > 590 nm) to the CMOS imaging camera (MICAM Ultima, Scimedia, Brainvision, Tokyo, Japan). The whole imaging system was connected through an I/O interface (Brainvision) to a PC controlling illumination, stimulation and data acquisition. The final pixel size was 2.5 μm with ×40 objective. Full-frame image acquisition was performed at 1 kHz. In this preparation, light scatter revealed by Point Spread Function was shown to be confined within a 10 μm diameter sphere[16]. Given maximal $\Delta F/F_0 \approx$ 1% and noise SEM ≈ ± 0.1% ($n = 12$ slices), the signal-to-noise (S/N) ratio was about 10 times ensuring a reliable measurement of peak response amplitude. Data were acquired and displayed by Brainvision software and signals were analyzed using custom-written routines. The analysis was performed by evaluating the maximum response of the VSD signal obtained at each tested frequency (10–500 Hz) normalized to the first peak response.

In these recordings, most of the VSD signal could be attributed to granule cells, as explained in detail in previous papers[15,16]. Briefly, given the numbers involved, the probability of recording activity from granule cells is 0.918, from Golgi cells is 0.002 and from a glomerulus is 0.08. Therefore, more than 90% of the signal had to derive from granule cells.

Long-term plasticity in VSDi recordings was induced as explained for SLM-2PM experiments and was measured as the variation between average peak amplitude before and after induction. LTP and LTD were identified as persistent positive or negative fluorescence variations and the stability of recordings was assessed by discarding pixels showing variations >±1σ from the control average.

The gain function was obtained by measuring the maximum amplitude of VSDi signals in each pixel. At each frequency, these amplitudes were normalized with respect to that of a single stimulus. The spatial analysis of granular layer filtering properties was obtained by generating and plotting the gain functions for all active pixels. A sigmoidal-shaped function $g(f)$ was fitted through the data:

$$g(f) = \frac{A_1 - A_2}{1 + (f/f_c)^p} + A_2 \qquad (2)$$

where $A_1$ and $A_2$ are the initial and final amplitude, $f_c$ is the cutoff frequency and $p$ is the order of the function (ORIGIN, Microcal Software Inc.).

The spatial distribution of the cutoff frequency was generated by plotting for each pixel the relative cutoff frequency. Based on the resulting maps, the number of pixels showing a given frequency was normalized to the total number of active pixels and averaged over different experiments.

**Granular layer circuit modeling and simulation**. Simulations of the cerebellar granular layer activity were performed using a realistic computational model that allows single-cell resolution of neuronal activity. The model, by incorporating new neuronal and synaptic mechanisms, provides a substantial update and extension of our previous model[30]. The model is written in NEURON + Python (v7.5, v2.7)[67,68] and is scalable, so that changes in network size lead automatically to rescaling the number of networks elements and their connectivity. In this way, the model achieves the scale required for simulation of salient operations in the cerebellar network[32], including ~400 × 10$^3$ neurons and ~2 × 10$^6$ synapses. Exploratory simulations were performed on a 72 cores/144 threads cluster (six blades with two Intel Xeon X5650 and 24 Gigabyte of DDR3 ram per blade). During simulations, the time step was fixed at 0.025 ms and the NEURON multi-split option was used to distribute computation corresponding to different cells over different cores (http://www.neuron.yale.edu/phpBB/)[67]. Massive simulations were run on the BlueGeneQ class supercomputer JUQUEEN at the Julich computational center within the Human Brain Project using resources allocated through the PRACE project cerebellum.

The granular layer network was constructed on the basis of detailed anatomical and functional information[8,69–76]) and using models of neurons and synapses including biophysical representations of membrane ionic channels and receptors[13,35,51,77–80]. These neuronal and synaptic models have been extensively validated using electrophysiological and imaging data. This allowed to develop a realistic network model, in which the large number of parameters is constrained to biology. In this bottom-up approach, the functional properties of the network emerge from the properties of constitutive elements and from their synaptic organization (e.g., see[81–83]). The availability of input spike patterns and neuronal

responses in vivo ([17,22,23,38,84,85] has allowed to simulate granular layer network dynamics under conditions representative of natural activity states. The network responses were validated by comparison with two-photon microscopy recordings[26] (this paper), as well as with previous local field potential[17], multi electrode array[15] and VSD[45] recordings of network activity.

**General properties and network architecture**. While being based on a previous network design[30], the size of the model was increased for representing a cerebellar slice with a thickness of ~150 μm (800 μm × 800 μm × 150 μm), with 384000 granule cells, 914 Golgi cells, and 29415 mossy fibers/glomeruli. Moreover, the cellular and synaptic mechanisms have been updated. The main extension were the introduction of excitatory connections from granule cells to Golgi cells through the ascending axon (aa)[86], of inhibitory connections among Golgi cells[87], of gap junctions between Golgi cells[88,89], of the orientation of the Golgi cell axon[70], of Golgi cell—granule cell wiring[90] (see also ref. [31]). The convergence/divergence ratios for each connection type have been update accordingly (see Supplementary Table s1).

Network connections were constructed using precise rules, yet allowing the number of connections and synaptic weights to show statistical variability (Gaussian distribution: mean = 1, s.d. = 0.4; see ref. [91]) and no systematic differences were observed using different seeds for parameter randomization. Background noise in the network was generated only by pacemaking in Golgi cells (see e.g., refs. [22,23,92]), since granule cells are silent at rest in slice preparations. Neurons and synapses were endowed with multiple receptor and ionic channel-based mechanisms, allowing an accurate representation of neuronal firing. The synapses were endowed with neurotransmitter diffusion mechanisms and with a representation of vesicle cycling, generating spillover and developing short-term facilitation and depression. Details on model construction and validation are given in Supplementary Note 1.

**Model of synaptic plasticity**. In the present work, long-term plasticity at the mossy fiber-granule cell synapse has been reproduced increasing (LTP) or decreasing (LTD) the release probability ($p$) according to the rules derived from the Bear–Lisman[40–42]. As explained previously, plasticity induction in granule cells is calcium-dependent[39], is bidirectional[28], and has a sliding threshold[93] compatible with the Bienenstock-Cooper-Munro rule[42]. Simulations of synaptic plasticity reproduced the applied experimental protocol:

1. Simulation of network activity before induction of plasticity; a 10-spikes burst at 50 Hz was delivered over 50 contiguous glomeruli. Release probability ($p$) was equal to 0.42 for all the mossy fiber-granule cell synapses.
2. Simulation of plasticity-induction protocol; 100 spikes at 100 Hz delivered over the same set of glomeruli. Single-cell voltage traces were recorded and used to estimate plasticity according to the Bear–Lisman approach (see below).
3. Simulation of network activity after induction of plasticity; same stimulation protocol as in 1. For each mossy fiber-granule cell synapses, $p$ changed according to 2.

Details on the synaptic plasticity model are given in Supplementary Note 1.

**Data analysis**. Data obtained from experiments and simulations were analyzed using routines custom written in Matlab (MathWorks, Natick, MA) and Python (v3.6). Activity response maps were used to calculate $E$, $I$, $E$–$I$ balance, gain and cutoff in simulations (starting from the average cell membrane potential) as when using the experimental data (starting from the average calcium or VSD signal). The 3D maps were generated using activity related parameters, including membrane voltage, spike discharge frequency, intracellular calcium and presynaptic release probability. The filtering properties of the response units were analyzed using a fast Fourier transform of the output signals.

**Statistics and reproducibility**. Data are reported as mean ± standard error of the mean (SEM) and statistical comparisons are done using paired and unpaired two-tailed Student's $t$ test, unless otherwise indicated (not significant at $p > 0.05$).

**Reporting summary**. Further information on research design is available in the Nature Research Reporting Summary linked to this article.

## Data availability

The data can be requested to the Authors and examples of SLM-2PM recordings are available on the Knowledge Graph (Human Brain Project). https://kg.ebrains.eu/search/live/minds/core/dataset/v1.0.0/4016ea85-aa19-470b-98b2-28448873ba03: XMPH-MTS[94]. Source data for the main figures is provided in Supplementary Data 1.

## Code availability

The code of the neuronal network model is publicly available on GitHub (https://github.com/dbbs-lab/Cerebellum-granular-layer-model-NEURON-Python).

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

## Acknowledgements

This research was supported by the European Union's Horizon 2020 Framework Programme for Research and Innovation under Specific Grant Agreement 720270 (Human Brain Project SGA1) and 785907 (Human Brain Project SGA2), with specific involvement of the Neuroinformatics Platform, Brain Simulation Platform, HPAC Platform and the Mouse Data (SP1) and Brain Modeling (SP6) subprojects. The work was also sponsored by the MNL project of the Centro Fermi (Rome, Italy) and the European supercomputing PRACE Project 2018184373.

## Author contributions

M.T. performed SLM-2PM experiments and analyzed the data; S.C. designed the models and performed the simulations; D.G. and J.M. performed VSDi experiments and analyzed the data; all authors contributed to paper writing and revision; E.D. coordinated research and wrote the final version of the paper.

## Competing interests

The authors declare no competing interests.
