## [Peer Review File · Communications Biology]

Reviewers' comments:

Reviewer #1 (Remarks to the Author):

The manuscript by S. Casali et. al. investigated the spatial filter tuning of the synaptic transmission at the cerebellum input stage by performing two-photon calcium imaging of the neurons in granular layer in vitro, and modeling the dynamics in the local microcircuit. In the experimental part, the authors first imaged the neuronal activity at the granular layer responding to mossy fiber stimulation. The neurons with increased and decreased neuronal activity was identified on a spatial map. This was then compared with the same experiment but with GABA-A receptors blocked. In the modeling and simulation part, the authors constructed a detailed model based on previously established results, and showed that the model could generate compatible results from the experiments. Using this model, the authors simulated the spatial distribution of LTP and LTD in the circuit, the synaptic spiking dynamics before and after plasticity induction, and finally the various spike filtering effects from different granule cells.

This is an interesting report of the spatiotemporal organization of LTP and LTD in local microcircuits, and of potential interest to the community. However, I have two major concerns:

1. While the results from the model agrees well with the calcium imaging, and the model itself has been validated with experimental results in the past, it is necessary to perform experiments to support the new predictions by the simulation, in particular the spike filtering effects. Adding these experiments can also further verify the model.
2. The authors mainly described the synaptic dynamics and filtering at the phenomenal level from the simulation. While this is valuable, it will add significant impact if the underlying biophysical mechanism could be studied and discussed from the model.

If the authors could address the above two concerns, I support the publication in Communications Biology.

Reviewer #2 (Remarks to the Author):

This is an excellent paper, describing two-photon recordings of cerebellar granule cells in acute slices, supported by modeling techniques, that test plasticity predictions formulated in Paul Dean's adaptive filter theory. Each recording collects data from about 300 active granule cells, allowing the authors to perform network experimentation and analysis. A core element of the study is to reveal the role of inhibition as well as a center-surround organization of plasticity probabilities with an LTP-dominated core and an LTD-dominated surround structure. The resulting 'Mexican hat' distribution adapts the information flow to the molecular layer of the cerebellar cortex and, ultimately, Purkinje cells.

The manuscript is outstanding. A few relatively minor revisions will further improve its impact:

p.6, l.132; and Fig. 2: the authors emphasize synaptic plasticity in this work, but it is unclear from their measures of calcium responses (in CaR-P/D) that these signals and their changes in plasticity are primarily synaptic in nature. Could not intrinsic excitability changes contribute, too, that have been demonstrated in this network (Armano et al., J. Neurosci. 20, 2000)? The major conclusions would probably stay the same, but the nature of these signals needs to be described with more accuracy.

p. 3, l.97: The authors write: 'Since multi-neuron maps were rather irregular, the spatial organization of neuronal activity was reconstructed by generating cumulative response maps from several recordings'. Are the authors referring to instability over time? If so, this observation is of interest on its own, and should be documented in detail. A similar phenomenon has been demonstrated in neocortex (Discroll et al., Cell 170, 2017) and should be cited here (if this is indeed a similar instability effect over time).

p. 13, l.244; and Fig 5: here, the authors show that overall depolarization was highest in the core, leading to a higher probability for LTP, and vice versa for LTD. As Mark Bear and Leon Cooper had shown (based on the original Bienenstock-Cooper-Munro model) LTP may result in a subsequently enhanced threshold for LTP, and a lower threshold for LTD (in Mark Bear's work, this phenomenon is CaMKII-dependent). Is there a similar sliding threshold phenomenon at mossy fiber-granule cell synapses? In the discussion, the authors might also refer to sliding thresholds for LTD and LTP at PF-Purkinje cell synapses (e.g. Piochon et al., PNAS 113, 2016), for comparison. As their manuscript will be particularly, but not exclusively, of interest to a cerebellar audience, it might be nice to add a paragraph in the discussion, talking about how their findings might affect plasticity at these PF synapses (for a critical update on plasticity rules at these synapses, see Tingley et al., J. Physiol. 597; 2019). Finally, I am not sure whether the term Lisman-Shouval model is established and really appropriate. To the understanding of the reviewer, this model was originating from the work of Bear and Cooper, with a critical involvement in one study by Shouval (cited). Maybe this doesn't matter that much, but it might be good to be more precise.

Reviewer #3 (Remarks to the Author):

In this MS, the authors used an optical method that they previously developed in order to study cerebellar granule cell activation following mossy fibers stimulation in acute cerebellar slices (Gandolfi, D., et al. Front Cell Neurosci 8, 92 690 (2014)). This method allows simultaneous calcium imaging in many granule cells in normal condition and when inhibition is blocked yielding a rough estimation of the inhibitory component in the microcircuit. They could show that groups of granule cells are differentially modulated by mossy fibers owing to a specific organization of the mossy fiber-Golgi cell feedforward inhibitory network. Using a simulation based on 3D modeling of the granule cell layer, they reproduced the observed features and identified new integration properties of the granule cell layer.

First, the authors should be commended for their work to develop new tools to study granule cell layer integration properties as it is still a central question to understand how the cerebellum compute incoming information. I found the model very precise and exhaustive, this part would certainly deserve more explanation in the main text instead of being in a supplementary file. However, I have yet some concerns about the biological data and the simulation:

- Related to E/I balance and the center-surround effect: E-I balanced is not exactly what is being used currently: here E is the total response (total $\Delta F/F$ when inhibition is ON) while I is the subtraction of the $\Delta F/F$ between control and Gabazine condition.

line 525 "Enorm and Inorm normalized by Enorm ($E-I = (Enorm - Inorm)/ Enorm$), where Enorm is the response intensity in control condition and Inorm is the response intensity variation after gabazine perfusion (both normalized to the maximum response), so that E-I values ranged from 1 (maximal excitation) to -1 (maximal inhibition)". Considering this definition, how Enorm-Inorm can be negative if Inorm is a fraction of Enorm as it is suggested? Please clarify. Is E a fraction of Enorm too?

Figure 1d does not clearly show center-surround signal, it is rather sparse, reflecting the 2P sparse illumination process and the patchy organization of mossy fiber inputs. Therefore this dataset hardly supports the claim for a center-surround effect. If any, could the "center-surround" effect be due to the extension of the Golgi cell axonal plexus?

How do the author differentiate between Golgi cells, granule cells and mossy fiber boutons in imaging data? Please provide analyses from individual granule cells, mossy fiber boutons and Golgi cells or alternatively a discussion explaining why granule cells selectively express FURA2. Please also show some individual maps, not only cumulative maps.

- Related to the simulation: in Figure 4a, please present the result of the stimulation as in Fig2c or 1d.

From supp: 1 Golgi cell targeting only 40 glomeruli seems an underestimation. Please give refs.

Mossy fibers give off collaterals in the same module/lobule yielding activation of several granule cell columns. Golgi cells may interact between columns and modify granule cell layer integration properties. How this could affect the simulation?

"By virtue of their extended dendritic fields, GoCs turned out to be connected beyond the volume occupied by the GrCs reached by the same MFs. Since the probability of getting connected to MFs was higher in the core of the bundle, the density of GrC excited by MFs decreased from core to periphery while the probability of GrC being inhibited remained high also in peripheral areas." Does this parameter explain the "center surround effect"? If so, you may want to modify this parameter in the simulation and evaluate its impact. Also, give refs suggesting such an organization.

The best argument to demonstrate recoding and pattern discrimination would be to show, using the simulation, that two groups of mossy fibers sharing fibers are well discriminated by a given group of granule cells.

-Please comment the recent experiments showing dense and localized granule cell activation (Giovannucci, A. et al. *Nat. Neurosci.* 20, 727–734 (2017); Wagner, M. J., Kim, T. H., Savall, J., Schnitzer, M. J. & Luo, L. *Nature* 544, 96–100 (2017); Valera, A. M. et al. *Elife* 5, e09862 (2016))

-How do this model relate to filtering by Fourier transformation by the different groups of granule cells (Straub, I. et al. *Elife* 9, 1–28 (2020))?

-A wealth of data demonstrated regional differences in the cerebellum (Cerminara, N. L., Lang, E. J., Sillitoe, R. V. & Apps, R. *Nat. Rev. Neurosci.* 16, 79–93 (2015). Which part of the cerebellum was used? Different region may have different operating rules.

This is an excellent paper, describing two-photon recordings of cerebellar granule cells in acute slices, supported by modeling techniques, that test plasticity predictions formulated in Paul Dean's adaptive filter theory. Each recording collects data from about 300 active granule cells, allowing the authors to perform network experimentation and analysis. A core element of the study is to reveal the role of inhibition as well as a center-surround organization of plasticity probabilities with an LTP-dominated core and an LTD-dominated surround structure. The resulting 'Mexican hat' distribution adapts the information flow to the molecular layer of the cerebellar cortex and, ultimately, Purkinje cells.

The manuscript is outstanding. A few relatively minor revisions will further improve its impact:

p.6, l.132; and Fig. 2: the authors emphasize synaptic plasticity in this work, but it is unclear from their measures of calcium responses (in CaR-P/D) that these signals and their changes in plasticity are primarily synaptic in nature. Could not intrinsic excitability changes contribute, too, that have been demonstrated in this network (Armano et al., J. Neurosci. 20, 2000)? The major conclusions would probably stay the same, but the nature of these signals needs to be described with more accuracy.

p. 3, l.97: The authors write: 'Since multi-neuron maps were rather irregular, the spatial organization of neuronal activity was reconstructed by generating cumulative response maps from several recordings'. Are the authors referring to instability over time? If so, this observation is of interest on its own, and should be documented in detail. A similar phenomenon has been demonstrated in neocortex (Discroll et al., Cell 170, 2017) and should be cited here (if this is indeed a similar instability effect over time).

p. 13, l.244; and Fig 5: here, the authors show that overall depolarization was highest in the core, leading to a higher probability for LTP, and vice versa for LTD. As Mark Bear and Leon Cooper had shown (based on the original Bienenstock-Cooper-Munro model) LTP may result in a subsequently enhanced threshold for LTP, and a lower threshold for LTD (in Mark Bear's work, this phenomenon is CaMKII-dependent). Is there a similar sliding threshold phenomenon at mossy fiber-granule cell synapses? In the discussion, the authors might also refer to sliding thresholds for LTD and LTP at PF-Purkinje cell synapses (e.g. Piochon et al., PNAS 113, 2016), for comparison. As their manuscript will be particularly, but not exclusively, of interest to a cerebellar audience, it might be nice to add a paragraph in the discussion, talking about how their findings might affect plasticity at these PF synapses (for a critical update on plasticity rules at these synapses, see Titley et al., J. Physiol. 597; 2019). Finally, I am not sure whether the term Lisman-Shouval model is established and really appropriate. To the understanding of the reviewer, this model was originating from the work of Bear and Cooper, with a critical involvement in one study by Shouval (cited). Maybe this doesn't matter that much, but it might be good to be more precise.

In this MS, the authors used an optical method that they previously developed in order to study cerebellar granule cell activation following mossy fibers stimulation in acute cerebellar slices (Gandolfi, D., et al. *Front Cell Neurosci* 8, 92 690 (2014)). This method allows simultaneous calcium imaging in many granule cells in normal condition and when inhibition is blocked yielding a rough estimation of the inhibitory component in the microcircuit. They could show that groups of granule cells are differentially modulated by mossy fibers owing to a specific organization of the mossy fiber-Golgi cell feedforward inhibitory network. Using a simulation based on 3D modeling of the granule cell layer, they reproduced the observed features and identified new integration properties of the granule cell layer.

First, the authors should be commended for their work to develop new tools to study granule cell layer integration properties as it is still a central question to understand how the cerebellum compute incoming information. I found the model very precise and exhaustive, this part would certainly deserve more explanation in the main text instead of being in a supplementary file. However, I have yet some concerns about the biological data and the simulation:

- Related to E/I balance and the center-surround effect: *E-I* balanced is not exactly what is being used currently: here *E* is the total response (total $\Delta F / F$ when inhibition is ON) while *I* is the subtraction of the $\Delta F / F$ between control and Gabazine condition.

line 525 "*Enorm* and *Inorm* normalized by *Enorm* ($E-I = (Enorm - Inorm) / Enorm$), where *Enorm* is the response intensity in control condition and *Inorm* is the response intensity variation after gabazine perfusion (both normalized to the maximum response), so that *E-I* values ranged from 1 (maximal excitation) to -1 (maximal inhibition)". Considering this definition, how *Enorm-Inorm* can be negative if *Inorm* is a fraction of *Enorm* as it is suggested? Please clarify. Is *E* a fraction of *Enorm* too?

Figure 1d does not clearly show center-surround signal, it is rather sparse, reflecting the 2P sparse illumination process and the patchy organization of mossy fiber inputs. Therefore this dataset hardly supports the claim for a center-surround effect. If any, could the "center-surround" effect be due to the extension of the Golgi cell axonal plexus?

How do the author differentiate between Golgi cells, granule cells and mossy fiber boutons in imaging data? Please provide analyses from individual granule cells, mossy fiber boutons and Golgi cells or alternatively a discussion explaining why granule cells selectively express FURA2. Please also show some individual maps, not only cumulative maps.

- Related to the simulation: in Figure 4a, please present the result of the stimulation as in Fig2c or 1d.

From supp: 1 Golgi cell targeting only 40 glomeruli seems an underestimation. Please give refs. Mossy fibers give off collaterals in the same module/lobule yielding activation of several granule cell columns. Golgi cells may interact between columns and modify granule cell layer integration properties. How this could affect the simulation?

"By virtue of their extended dendritic fields, GoCs turned out to be connected beyond the volume occupied by the GrCs reached by the same MFs. Since the probability of getting connected to MFs was higher in the core of the bundle, the density of GrC excited by MFs decreased from core to periphery while the probability of GrC being inhibited remained high also in peripheral areas." Does this parameter explain the "center surround effect"? If so, you may want to modify this parameter in the simulation and evaluate its impact. Also, give refs suggesting such an organization.

The best argument to demonstrate recoding and pattern discrimination would be to show, using the simulation, that two groups of mossy fibers sharing fibers are well discriminated by a given group of granule cells.

-Please comment the recent experiments showing dense and localized granule cell activation (Giovannucci, A. *et al. Nat. Neurosci.* 20, 727–734 (2017); Wagner, M. J., Kim, T. H., Savall, J., Schnitzer, M. J. & Luo, L. *Nature* 544, 96–100 (2017); Valera, A. M. *et al. Elife* 5, e09862 (2016))

-How do this model relate to filtering by Fourier transformation by the different groups of granule cells (Straub, I. *et al. Elife* 9, 1–28 (2020))?

-A wealth of data demonstrated regional differences in the cerebellum (Cerminara, N. L., Lang, E. J., Sillitoe, R. V. & Apps, R. *Nat. Rev. Neurosci.* 16, 79–93 (2015). Which part of the cerebellum was used? Different region may have different operating rules.

Reviewers' comments:

Reviewer #1 (Remarks to the Author):

The manuscript by S. Casali et. al. investigated the spatial filter tuning of the synaptic transmission at the cerebellum input stage by performing two-photon calcium imaging of the neurons in granular layer in vitro, and modeling the dynamics in the local microcircuit. In the experimental part, the authors first imaged the neuronal activity at the granular layer responding to mossy fiber stimulation. The neurons with increased and decreased neuronal activity was identified on a spatial map. This was then compared with the same experiment but with GABA-A receptors blocked. In the modeling and simulation part, the authors constructed a detailed model based on previously established results, and showed that the model could generate compatible results from the experiments. Using this model, the authors simulated the spatial distribution of LTP and LTD in the circuit, the synaptic spiking dynamics before and after plasticity induction, and finally the various spike filtering effects from different granule cells.

This is an interesting report of the spatiotemporal organization of LTP and LTD in local microcircuits, and of potential interest to the community. However, I have two major concerns:

1. While the results from the model agrees well with the calcium imaging, and the model itself has been validated with experimental results in the past, it is necessary to perform experiments to support the new predictions by the simulation, in particular the spike filtering effects. Adding these experiments can also further verify the model.

We thank the Reviewer for the insightful comments and appreciate the proposal for validating model predictions. Indeed, we have performed new recordings generating an independent data set with a different technique (voltage sensitive dye imaging, VSDi). This has involved the collaboration of two additional researchers, who are now in the Author list. Since the VSDi signal reflects the integral of all membrane potential changes occurring in a pixel averaged over the recording time, VSDi recordings can be directly compared to simulations. The experiments involved mossy fiber stimulation at different frequencies, before and after the induction of long-term synaptic plasticity. Model simulations were designed and run equivalently. Amazingly, VSDi signal patterning closely matched modeling predictions over the space and frequency domains providing a high-level validation to the model. A new figure (Fig. 7) has been added and the text updated accordingly.

2. The authors mainly described the synaptic dynamics and filtering at the phenomenal level from the simulation. While this is valuable, it will add significant impact if the underlying biophysical mechanism could be studied and discussed from the model.

Indeed, the model provides a unique tool for understanding the mechanisms of network processing. Previous experimental observations using VSD recordings revealed that NMDA receptor-mediated depolarization is key to explain gain changes in the granular layer (Mapelli et al., 2010; Solinas et al., 2010). Interestingly, the model predicted that gain reflected the balance between activation of NMDA and AMPA receptors vs. GABA receptors. This balance was higher in the core than in the periphery of the unit. And since LTP was due to enhanced neurotransmitter release and raised both NMDA and AMPA receptor activation, it especially improved signal transmission in the core of the unit, while LTD sorted the opposite effect. This reasoning was extended to the frequency domain, considering that NMDA receptors have slow voltage-dependent activation enhancing low-frequency transmission. Therefore, the model provided the keys for a mechanistic explanation of unit's functioning and regulation.

If the authors could address the above two concerns, I support the publication in Communications

Biology.

Reviewer #2 (Remarks to the Author):

This is an excellent paper, describing two-photon recordings of cerebellar granule cells in acute slices, supported by modeling techniques, that test plasticity predictions formulated in Paul Dean's adaptive filter theory. Each recording collects data from about 300 active granule cells, allowing the authors to perform network experimentation and analysis. A core element of the study is to reveal the role of inhibition as well as a center-surround organization of plasticity probabilities with an LTP-dominated core and an LTD-dominated surround structure. The resulting 'Mexican hat' distribution adapts the information flow to the molecular layer of the cerebellar cortex and, ultimately, Purkinje cells.

The manuscript is outstanding. A few relatively minor revisions will further improve its impact:

We thank the Reviewer for the appreciation and the insightful comments. We have followed the recommendations and emended the paper accordingly.

p.6, l.132; and Fig. 2: the authors emphasize synaptic plasticity in this work, but it is unclear from their measures of calcium responses (in CaR-P/D) that these signals and their changes in plasticity are primarily synaptic in nature. Could not intrinsic excitability changes contribute, too, that have been demonstrated in this network (Armano et al., J. Neurosci. 20, 2000)? The major conclusions would probably stay the same, but the nature of these signals needs to be described with more accuracy.

Yes indeed, this possibility has been mentioned in the paper. In principle, if the changes in intrinsic excitability were bidirectional, they would enhance the difference between the LTP and LTD areas. However, the description provided by Armano et al. (2020) and the subsequent elaboration by Nieuwenhuis et al. (2006) were only about LTP. We thought it would be preliminary to assume bidirectional induction of non-synaptic plasticity in cerebellar granule cells and preferred not to anticipate it in through model simulations. But we appreciate the suggestion and will surely consider the experimental investigation and modeling of the of the issue in the near future, as soon as the induction rule will be clarified. We have added a comment in the discussion.

p. 3, l.97: The authors write: 'Since multi-neuron maps were rather irregular, the spatial organization of neuronal activity was reconstructed by generating cumulative response maps from several recordings'. Are the authors referring to instability over time? If so, this observation is of interest on its own, and should be documented in detail. A similar phenomenon has been demonstrated in neocortex (Discroll et al., Cell 170, 2017) and should be cited here (if this is indeed a similar instability effect over time).

This comment opens a very interesting perspective that has been added to the discussion. We observed variations from map to map both as a consequence of local variability in neuronal wiring across regions and as a consequence of plasticity in the same map. It is interesting to speculate that this variability might be the substrate for the map changes occurring along daily life in freely behaving mice as shown by Discroll in neocortex. We have added a comment in the discussion.

p. 13, l.244; and Fig 5: here, the authors show that overall depolarization was highest in the core, leading to a higher probability for LTP, and vice versa for LTD. As Mark Bear and Leon Cooper had shown (based on the original Bienenstock-Cooper-Munro model) LTP may result in a subsequently enhanced threshold for LTP, and a lower threshold for LTD (in Mark Bear's work, this phenomenon is CaMKII-dependent). Is there a similar sliding threshold phenomenon at mossy fiber-granule cell synapses? In the discussion, the authors might also refer to sliding thresholds for LTD and LTP at PF-Purkinje cell synapses (e.g. Piochon et al., PNAS 113, 2016), for comparison. As their

manuscript will be particularly, but not exclusively, of interest to a cerebellar audience, it might be nice to add a paragraph in the discussion, talking about how their findings might affect plasticity at these PF synapses (for a critical update on plasticity rules at these synapses, see Titley et al., J. Physiol. 597; 2019). Finally, I am not sure whether the term Lisman-Shouval model is established and really appropriate. To the understanding of the reviewer, this model was originating from the work of Bear and Cooper, with a critical involvement in one study by Shouval (cited). Maybe this doesn't matter that much, but it might be good to be more precise.

Indeed, a more appropriate name to indicate the plasticity rule that we have applied, could be "Bear-Lisman". The paper from Cooper and Bear is now mentioned and indeed the LTP/LTD neutral point of mossy fiber-granule cell plasticity is sliding as in the classical BCM formulation (this was demonstrated by Prestori et al., 2013). Lisman remains in the name because of his original formulation on bidirectional calcium-dependent induction of plasticity. The impact of granular layer pre-processing on plasticity at the parallel fiber-Purkinje cell synapse has been commented.

Reviewer #3 (Remarks to the Author):

In this MS, the authors used an optical method that they previously developed in order to study cerebellar granule cell activation following mossy fibers stimulation in acute cerebellar slices (Gandolfi, D., et al. Front Cell Neurosci 8, 92 690 (2014)). This method allows simultaneous calcium imaging in many granule cells in normal condition and when inhibition is blocked yielding a rough estimation of the inhibitory component in the microcircuit. They could show that groups of granule cells are differentially modulated by mossy fibers owing to a specific organization of the mossy fiber-Golgi cell feedforward inhibitory network. Using a simulation based on 3D modeling of the granule cell layer, they reproduced the observed features and identified new integration properties of the granule cell layer.

First, the authors should be commended for their work to develop new tools to study granule cell layer integration properties as it is still a central question to understand how the cerebellum compute incoming information. I found the model very precise and exhaustive, this part would certainly deserve more explanation in the main text instead of being in a supplementary file. However, I have yet some concerns about the biological data and the simulation:

We thank the Reviewer for the appreciation, the precise analysis of our work and the insightful comments. We have carefully considered the suggestions and emended the paper accordingly. About modeling, we have added a figure in the main text (Fig.3).

- Related to E/I balance and the center-surround effect: E-I balanced is not exactly what is being used currently: here E is the total response (total $\Delta F/F$ when inhibition is ON) while I is the subtraction of the $\Delta F/F$ between control and Gabazine condition.

line 525 "Enorm and Inorm normalized by Enorm ($E-I = (E_{norm} - I_{norm})/ E_{norm}$), where E_{norm} is the response intensity in control condition and I_{norm} is the response intensity variation after gabazine perfusion (both normalized to the maximum response), so that E-I values ranged from 1 (maximal excitation) to -1 (maximal inhibition)". Considering this definition, how $E_{norm}-I_{norm}$ can be negative if I_{norm} is a fraction of E_{norm} as it is suggested? Please clarify. Is E a fraction of E_{norm} too?

We have clarified this point by editing the text in Methods as follows:

“

As explained previously (Mapelli et. al, 2007; Soda et al., 2019), the excitatory-inhibitory (E-I) balance maps were computed for each experiment as

$$E-I = (E_{norm} - I_{norm})/ E_{norm},$$

where E_{norm} is the response intensity in control condition and I_{norm} is the response intensity variation after gabazine perfusion (both normalized to their maximum response), so that E-I values ranged from 1 (maximal excitation) to -1 (maximal inhibition).

“

Just to make a numerical example, consider that E is the response when inhibition is active and G is the response when inhibition is blocked (i.e. during Gabazine perfusion). I is obtained as $I=G-E$ (the measure can be $\Delta F/F_0$ or any other experimental variable).

Example 1 (a cell with a minor variation during Gabazine perfusion): $E_{\Delta F/F_0}=0.37$ and $G_{\Delta F/F_0} = 0.38$, then $I_{\Delta F/F_0}=G_{\Delta F/F_0}-E_{\Delta F/F_0} = 0.01$ and finally $(E_{\Delta F/F_0}-I_{\Delta F/F_0})/E_{\Delta F/F_0} = 0.97$.

Example 2 (a cell with a major variation during Gabazine perfusion): $E_{\Delta F/F_0}=0.43$ and $G_{\Delta F/F_0} = 1.03$, then $I_{\Delta F/F_0}=G_{\Delta F/F_0}-E_{\Delta F/F_0} = 0.6$ and finally $(E_{\Delta F/F_0}-I_{\Delta F/F_0})/E_{\Delta F/F_0} = -0.39$.

This simple method allows to discriminate between cells (or pixels or voxels), in which either inhibition or excitation is prevalent.

Figure 1d does not clearly show center-surround signal, it is rather sparse, reflecting the 2P sparse illumination process and the patchy organization of mossy fiber inputs. Therefore this dataset hardly supports the claim for a center-surround effect. If any, could the "center-surround" effect be due to the extension of the Golgi cell axonal plexus?

Following the reviewer's comment on the SLM-2PM maps, we now mention that "These cumulative maps *were consistent* with the center/surround (C/S) organization observed using local field potential recordings and voltage-sensitive dye imaging". It should be noted that SLM-2PM provides a fine-grain representation of neuronal activity in multiple single neurons. Therefore, the maps are conceivably noisier than those generated with MEA or VSD, where the signal is filtered by electrical diffusion or light diffraction and signal sampling is not as sparse as with SLM-2PM. The explanation of the center-surround effect was reported in Mapelli et al. (2007) and corroborated by circuit simulations in Solinas et al. (2010). We have added a panel in Figure 6 that shows, at the mechanistic level, how the center-surround effect can be generated by the different E/I balance in center and periphery of the units. Which, in turn, descends from lateral inhibition provided by the extended Golgi cell axonal plexus (see also below).

How do the author differentiate between Golgi cells, granule cells and mossy fiber boutons in imaging data? Please provide analyses from individual granule cells, mossy fiber boutons and Golgi cells or alternatively a discussion explaining why granule cells selectively express FURA2.

The way cells were selected is explained in the original methodological paper (Gandolfi et al., 2014). The same procedure and arguments are valid here and reported in the section on Methods. Each experimental session started with the acquisition of a two-photon image of the granular layer: granule cells were identified by their soma shape and size, which differentiate them from the mossy fiber boutons and the Golgi cells. Using this procedure, the possibility of getting neural structures other than granule cells was remote (Gandolfi et al., 2014). It should be noted that, even without selection of the neural structures, given the numbers involved, the probability of getting a Golgi cell or a glomerulus would be negligible (0,002 and 0,08, respectively). This issue is now mentioned in Methods.

Please also show some individual maps, not only cumulative maps.

We have added to Supplementary Material Fig. s1 showing examples of individual excitatory-inhibitory maps reconstructed from single experiments.

- Related to the simulation: in Figure 4a, please present the result of the stimulation as in Fig2c or 1d.

The requested representation is now added as Fig. 4c

From supp: 1 Golgi cell targeting only 40 glomeruli seems an underestimation. Please give refs.

To our knowledge, there is no direct assessment of the number of glomeruli targeted by a single Golgi cell and the number of 40 is the result of calculations. We have assumed that only one Golgi cell axon enters a glomerulus, forming inhibitory synapses on all the connected granule cell dendrites, and that a Golgi cell axon entering a glomerulus cannot access the neighboring glomeruli if they share dendrites coming from the same granule cells of the first one. This prevents a granule cell from being inhibited more than once by the same Golgi cell, a case has never been observed experimentally (Mapelli et al., 2014). Under this assumption, each Golgi cell can inhibit as many as 40 different glomeruli and a total of about 2000 granule cells, accounting for the 1:430 Golgi:granule cell ratio and the convergence and divergence ratios reported from anatomy (Korbo et al., 1993). This number of 40 is also akin with recent calculations suggesting that each Golgi cell

receives excitatory inputs from about 40 mossy fibers on basal dendrites (Kanichay and Silver, 2008). Clearly, the geometry of Golgi cell inhibition could be updated (see e.g. a recent paper by Tabuchi et al., 2019) and the assumption of 1 Golgi cell per glomerulus relaxed, but we feel this is out of the scope of the present work and of the scale of simulations (see below).

Mossy fibers give off collaterals in the same module/lobule yielding activation of several granule cell columns. Golgi cells may interact between columns and modify granule cell layer integration properties. How this could affect the simulation?

Simulations were carried out on an $800 \times 800 \times 150 \mu\text{m}^3$ portion of the cerebellar granular layer. Therefore, it was not needed here to simulate high-level structural organization across modules. However, we agree this is quite relevant in the context of meso-scale network interactions and could be done through specific simulations using an extended version of this same network (this is something we are actually planning to do soon, see below). As for the present case, in line with the target of the paper, we have given a hint on how different components of the Golgi cell network might regulate the LTP/LTD balance in Supplemental Material.

"By virtue of their extended dendritic fields, GoCs turned out to be connected beyond the volume occupied by the GrCs reached by the same MFs. Since the probability of getting connected to MFs was higher in the core of the bundle, the density of GrC excited by MFs decreased from core to periphery while the probability of GrC being inhibited remained high also in peripheral areas." Does this parameter explain the "center surround effect"? If so, you may want to modify this parameter in the simulation and evaluate its impact. Also, give refs suggesting such an organization.

Yes, exactly, this is the explanation of the center-surround effect as previously given in Mapelli et al. (2007) for MEA recordings, Mapelli et al. (2010) and Soda et al. (2019) for VSD recordings, Gandolfi et al. (2014) for 2PM recordings, Solinas et al. (2010) for computational simulations. Again, it would be very nice to use the model in order to understand in depth the impact of connectivity. In order to do so, we are elaborating a new network codebase (paper in preparation), in which connectivity parameters can be easily changed in a scalable fashion (briefly, in the new code, connectivity is related to geometry of neural processes through the probability of forming synapses between crossing segments). In the present paper, the model is just configured to address the impact of parameters related to plasticity on spatial adaptive filtering.

The best argument to demonstrate recoding and pattern discrimination would be to show, using the simulation, that two groups of mossy fibers sharing fibers are well discriminated by a given group of granule cells.

To date, it has been hard to face the main Marr's theoretical postulate, according to which granule cells can separate incoming patterns. Marr did not make any assumption either on the coding properties of granular layer neurons or on the input mossy fiber patterns, and not even specified the clustering of granule cells that take part to local computations. We did not touch the issue of spatial pattern separation explicitly in the first version of the paper, but this could be done, and we thank the reviewer for the suggestion. Thus, in the present revision, we have extended the simulations by activating 2 response units simultaneously (Fig.8b), each unit carrying a different frequency pattern. Unit 1 receives a 15 Hz pattern, unit 2 a 30 Hz pattern. Unit 1 and 2 overlap by 35% (26 out of 74 glomeruli are in common) of their mossy fibers. Indeed, the simulations showed that the output reflected the frequencies of the two inputs, almost irrespective of the overlap (Fig. 8b). Therefore, the network was able to spatially separate patterns at the output despite their partial overlapping at the input.

-Please comment the recent experiments showing dense and localized granule cell activation

(**Giovannucci**, A. et al. Nat. Neurosci. 20, 727–734 (2017); **Wagner**, M. J., Kim, T. H., Savall, J., Schnitzer, M. J. & Luo, L. Nature 544, 96–100 (2017); Valera, A. M. et al. Elife 5, e09862 (2016))
-How do this model relate to filtering by Fourier transformation by the different groups of granule cells (Straub, I. et al. Elife 9, 1–28 (2020))?
-A wealth of data demonstrated regional differences in the cerebellum (**Cerminara**, N. L., Lang, E. J., Sillitoe, R. V. & Apps, R. Nat. Rev. Neurosci. 16, 79–93 (2015). Which part of the cerebellum was used? Different region may have different operating rules.

All these references have now been considered in the discussion. We believe the “units” revealed here represent the substrate for the dense clusters revealed by Diwakar and more recently seen in action by Giovannucci and Wagner. We also argue that the units may provide the substrate for the Fourier transformation discovered by Straub, very much in line with the adaptive filter model of Dean and Porrill.

The rules that we have reported for the units concern the cerebellar vermis, mostly laminae V-VI. The group of Apps has nicely shown differences among cerebellar regions and this may reverberate on the way the units process incoming signals. By using the new model codebase that we have mentioned above we are just going in this direction. We are already tailoring local connectivity to atlases and literature data in order to analyze the local differences but, as said, this will require a technological development that is not yet available in the present version of the model.

REVIEWERS' COMMENTS:

Reviewer #1 (Remarks to the Author):

The authors have addressed my previous comments. The manuscript is comprehensive. The simulations and experiments match well and support the conclusion. I support its publication in Communications Biology.

Reviewer #2 (Remarks to the Author):

All my previous concerns have been appropriately addressed.